# Circulating extracellular vesicle microRNAs mediate immune modulation of social behavior in male mice

Ken Matoba[1,9], Eisuke Dohi [1,2,7,9], Phoebe A. Garcia [1,9], Jose Francis-Oliveira [1,9], Mirmohammadali Mirramezanializamini[1], Inssaf Berkiks[1], Frida Anguiano[1], Jana H. Badrani [1], Oluwaseun Fatoba[1], Eric Y. Choi [2], Julia See[2], Md. Sorwer Alam Parvez [1], Takahiro Kochi[1], Norimichi Ito[1], Rei Mitani[2], Indigo V. L. Rose [2], Takashi Imai[2], David K. Crossman [3], Mikhail V. Pletnikov[2,8], Kenneth W. Witwer[4], Minae Niwa [1,2,5,6] & Shin-ichi Kano [1,2,5] ✉

Extracellular vesicles (EVs) are cell-derived small membrane vesicles and circulate throughout the body, but the impact of circulating EVs on brain function and behavior remains elusive. Here, we report that wild-type (WT) mouse blood, particularly EVs, increases sociability in socially impaired immunodeficient $Rag1^{-/-}$ male mice, mimicking the effects of WT T cell transfer. These EVs are localized to neurons and regulate PKCε expression, $GABA_A$ receptor synaptic localization, and inhibitory postsynaptic signaling in prefrontal cortex (PFC) pyramidal neurons. Injection of $Rag1^{-/-}$ EVs supplemented with miR-23a-3p and miR-103-3p enhances synaptic function and sociability in $Rag1^{-/-}$ mice. T cells secrete miR-23a-3p via EVs, and $Mir23a^{-/-}$ T cells fail to increase sociability. Similar beneficial effects of WT blood EVs are observed in additional mouse models with sociability deficits, such as $Cntnap2^{-/-}$ and $Shank3^{-/-}$ mice. These findings uncover the role of EV miRNAs in mediating immune modulation of synaptic function and social behavior, revealing a non-canonical molecular pathway for immune-neuron communication.

Factors circulating in the blood can influence brain function and behavior in rodents. An emerging concept in aging research is that the transfusion of young rodent blood into aged rodents enhances memory function[1,2]; however, it is unclear whether normal blood components ameliorate other behavioral deficits in a non-aging context. Accumulating evidence supports the requirement of lymphocytes, such as T cells, for brain function and behavior. Notably, immunodeficient mice, lacking T and B cells, display deficits in various

[1]Department of Psychiatry and Behavioral Neurobiology, University of Alabama at Birmingham Heersink School of Medicine, Birmingham, AL, USA. [2]Department of Psychiatry and Behavioral Sciences, Johns Hopkins University School of Medicine, Baltimore, MD, USA. [3]Department of Genetics, University of Alabama at Birmingham Heersink School of Medicine, Birmingham, AL, USA. [4]Departments of Molecular and Comparative Pathobiology and Neurology, and The Richman Family Precision Medicine Center of Excellence in Alzheimer's Disease, Johns Hopkins University School of Medicine, Baltimore, MD, USA. [5]Department of Neurobiology, University of Alabama at Birmingham Heersink School of Medicine, Birmingham, AL, USA. [6]Department of Biomedical Engineering, University of Alabama at Birmingham School of Engineering, Birmingham, AL, USA. [7]Present address: National Institute of Neuroscience, National Center of Neurology and Psychiatry, Tokyo, Japan. [8]Present address: Department of Physiology and Biophysics, Jacobs School of Medicine and Biomedical Sciences, SUNY University at Buffalo, Buffalo, NY, USA. [9]These authors contributed equally: Ken Matoba, Eisuke Dohi, Phoebe A. Garcia, Jose Francis-Oliveira. ✉e-mail: sikano@uabmc.edu

behaviors, such as learning and memory, anxiety-related behaviors, and social behaviors[3–6]. Although T cell-derived cytokines, such as interferon-gamma (IFN-γ), interleukin-4 (IL-4), and IL-17, have been shown to influence these behaviors[4,5,7], additional factors may be involved in immune regulation of brain function and behavior. Indeed, lymphocytes have been shown to affect gut microbiota-derived metabolites, glucocorticoids, amino acids, and extracellular micro-RNAs (miRNAs) in the blood[6,8–10].

Here, to further address the influence of circulating blood components on brain function and behavior, we examined the effects of blood transfusion on social behaviors using immunodeficient mice. We discovered that intravenous injection of extracellular vesicles (EVs) from wild-type (WT) mouse blood, as well as WT mouse serum transfusion, improved sociability but not social novelty preference in immunodeficient *Rag1*[−/−] mice. WT blood EVs modulate inhibitory postsynaptic currents (IPSCs) in the medial prefrontal cortex (mPFC) via miRNA-mediated control of gene expression. We identified T cells as the source of EVs and miRNAs that influence synaptic function and sociability. Notably, WT blood EVs also increased sociability and IPSCs in two additional mouse models, *Cntnap2*[−/−] and *Shank3*[−/−] mice, highlighting a potential shared mechanism. Therefore, we have identified a notable role of circulating blood EVs on brain function and behavior.

## Results

### Intravenous injection of WT serum enhances sociability in immunodeficient *Rag1*[−/−] mice

Previous studies reported that immunodeficient *scid* and *Rag2*[−/−] mice, lacking T and B cells, showed sociability deficits in the three-chamber social interaction test[5,11]. Consistent with these findings, we observed that *Rag1*[−/−] mice also exhibited sociability deficits in this assay (Fig. 1a, b), despite normal performance in olfactory function and open field exploration (Supplementary Fig. 1a, b). To determine whether immune cell reconstitution could beneficially impact sociability, we performed adoptive transfer of splenocytes or purified T cells from WT mice into *Rag1*[−/−] mice. Both transfers significantly increased sociability in *Rag1*[−/−] mice (Fig. 1c, d, and Supplementary Fig. 2a, b). This suggests that the behavioral deficits were due to the absence of T cells, as Rag1 protein was undetectable in the brain during prenatal development, early postnatal stages, and adulthood (Supplementary Fig. 1c, d). Intravenous injection of serum from WT mice also increased sociability in *Rag1*[−/−] mice (Fig. 1e). Although circulating cytokines such as IFN-γ and IL-17 were shown to modulate social behavior[5,7], their concentrations in WT serum were below the detection limit of the enzyme-linked immunosorbent assay (Supplementary Fig. 2c). These findings suggest that other circulating factors in WT blood may mediate the observed behavioral improvement. Therefore, we sought to identify additional blood components that might regulate sociability.

### T cell-dependent improvement in sociability by *Rag1*[−/−] mice is associated with changes in EV-associated miRNAs in the blood

Extracellular vesicles (EVs) are nanosized, membrane-enclosed vesicles that carry diverse cell-derived molecular cargo, such as RNAs, proteins, lipids, and carbohydrates[12–14]. Secreted by virtually all cell types, EVs are involved in waste disposal, antigen presentation, and intercellular communication[12–17]. Once released into circulation, EVs can reach distant organs and modulate their function[18,19]. Circulating EV-associated molecules, particularly those of brain origin, have emerged as promising biomarkers in neurological and psychiatric disorders, including Alzheimer's disease, psychotic disorders, depression, and autism spectrum disorders (ASDs)[20–32]. Moreover, several studies suggest that circulating EVs negatively impact brain function and behavior under disease conditions[23,32,33]. Nevertheless, whether and how circulating EVs and their molecular cargo positively influence brain function and behavior remains unknown.

To determine whether EVs contribute to the behavioral improvement by WT serum, we collected blood EVs (hereafter bEVs) from WT and *Rag1*[−/−] sera. Although the morphology, size distribution, and concentration of bEVs were comparable between WT and *Rag1*[−/−] mice, we observed reduced expression of an EV marker CD9 protein in *Rag1*[−/−] bEVs (Supplementary Fig. 3a–f). CD3ε protein in bEVs, likely derived from T cells, was also significantly diminished in *Rag1*[−/−] mice (Supplementary Fig. 3f). Thus, *Rag1*[−/−] bEVs were qualitatively altered and lack T cell-derived EVs. Adoptive transfer of WT T cells, which increased sociability (Fig. 1d), also increased the expression levels of CD3ε protein and miRNAs in *Rag1*[−/−] bEVs (Fig. 1f, g, and Supplementary Data 1). Notably, the differentially expressed miRNAs significantly overlapped with those altered in multiple mouse models exhibiting sociability deficits (Fig. 1h). Predicted target genes of these miRNAs were enriched for pathways related to neuronal and synaptic function (Fig. 1i and Supplementary Data 2–4). These findings suggest that T cell-derived EVs and their miRNAs cargo may modulate sociability by influencing synaptic gene expression in the brain.

### WT blood EV transfer improves sociability in *Rag1*[−/−] mice without affecting social novelty preference

To directly assess the impact of circulating EVs on social behavior, we intravenously injected WT bEVs into *Rag1*[−/−] mice (Fig. 2a). WT bEV injection did not change the numbers of peripheral T or B cells (Supplementary Fig. 4) but significantly increased sociability in *Rag1*[−/−] mice (Fig. 2b, and Supplementary Fig. 5a). Notably, WT bEVs also increased sociability in *Cntnap2*[−/−] and *Shank3*[−/−] mice, two established mouse models of sociability deficits (Fig. 2c, d). In contrast, WT bEVs had no effect on sociability in WT mice (Fig. 2e), excluding the possibility that a mere increase in circulating EVs may enhance sociability. *Rag1*[−/−] mice also exhibited deficits in social novelty preference in the three-chamber social interaction test (Fig. 2f, g). However, WT bEVs did not improve social novelty preference in these mice (Fig. 2h and Supplementary Fig. 5b), indicating a selective effect on neuronal function underlying sociability. Consistently, the transfer of either WT sera, splenocytes, or T cells also failed to increase social novelty preference in *Rag1*[−/−] mice (Supplementary Fig. 5c, d). Together, these findings demonstrate that WT bEVs selectively improve sociability, but not social novelty preference, in *Rag1*[−/−] mice.

### Circulating bEVs reach the brain parenchyma and co-localize with neurons and microglia in *Rag1*[−/−] mice

To gain insight into the mechanisms underlying sociability improvement by WT bEVs, we examined their distribution in the brain following intravenous injection. WT bEVs were first labeled with a lipophilic dye, PKH26, and intravenously injected into *Rag1*[−/−] mice. As early as 1 h post injection, labeled bEVs were detected throughout the brain, including the medial prefrontal cortex (mPFC), the hippocampus (HC), and the cerebellum (CB) (Fig. 3a, b). As lipid dye labeling may be promiscuous and generate non-EV micelles[34], we validated these findings using bEVs collected from mTmG mice, which express a membrane-anchored tdTomato fluorescent protein[35]. Consistent with the PKH26 labeling, tdTomato-positive bEVs were detected in the mPFC of *Rag1*[−/−] mice (Fig. 3c). Cell-type-specific analysis revealed that WT bEVs co-localized with neurons and microglia, but not with astrocytes or oligodendrocytes (Fig. 3d, e, and Supplementary Fig. 6a, b), suggesting that these cell types may be primary recipients of circulating EVs in the brain.

### WT bEVs improve sociability by attenuating hyperexcitability in the mPFC

The mPFC plays a central role in the control of sociability behaviors[36–39]; hence, we determined its involvement in sociability deficits and their improvement by WT bEVs in *Rag1*[−/−] mice. We first examined neuronal activities using c-Fos protein expression, an

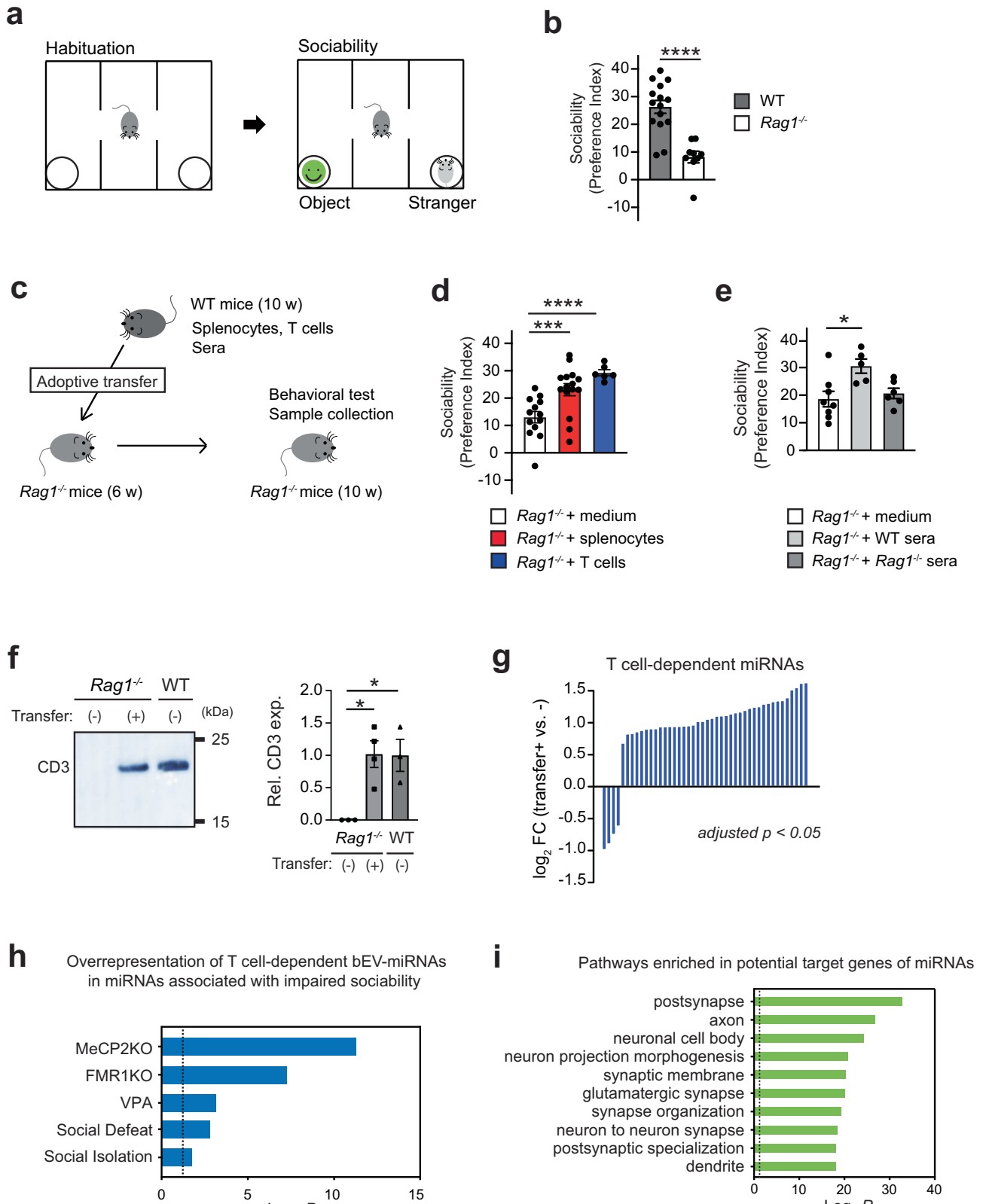

indicator of neuronal activity. c-Fos expression in the mPFC was increased in *Rag1*⁻/⁻ mice, mostly in CaMKIIα⁺ excitatory neurons, compared to WT mice (Fig. 4a, b, and Supplementary Fig. 7a, b). To address the causal role of the increased activity of CaMKIIα⁺ neurons in sociability deficits in *Rag1*⁻/⁻ mice, we then conducted chemogenetic inhibition using an inhibitory DREADD (designer receptors exclusively

activated by designer drugs), hM4D(Gi)[40]. Expression of hM4D(Gi) via adeno-associated virus (AAV) vectors [AAV-CaMKIIα-hM4D(Gi)-mCherry] in the mPFC CaMKIIα⁺ neurons increased sociability in *Rag1*⁻/⁻ mice upon the intravenous injection of a DREADD agonist, clozapine *N*-oxide (CNO) (Fig. 4c, and Supplementary Fig. 8a–d). Notably, intravenous injection of WT bEVs reduced the mPFC c-Fos expression in

**Fig. 1 | Serum injection as well as T cell transfer from WT mice improves sociability in _Rag1⁻/⁻_ mice. a** Outline of the three-chamber sociability test. **b** Sociability in the three-chamber social interaction test by WT and _Rag1⁻/⁻_ mice. WT mice, n = 15; _Rag1⁻/⁻_ mice, n = 9. **c** Experimental outline of adoptive transfer/ serum injection and subsequent behavioral assay. **d** Adoptive transfer of WT mouse splenocytes or T cells into _Rag1⁻/⁻_ mice increased their sociability. _Rag1⁻/⁻_ mice + medium, n = 13; _Rag1⁻/⁻_ mice + WT splenocytes, n = 15; and _Rag1⁻/⁻_ mice + WT T cells, n = 6. **e** Intravenous injection of sera from WT mice, but not from _Rag1⁻/⁻_ mice, into _Rag1⁻/⁻_ mice increased their sociability. _Rag1⁻/⁻_ mice + medium, n = 8; _Rag1⁻/⁻_ mice + WT sera, n = 5; and _Rag1⁻/⁻_ mice + _Rag1⁻/⁻_ sera, n = 6. **f** Presence of CD3⁺ EVs in WT and _Rag1⁻/⁻_ samples. Left, representative Western blot images showing the recovery of CD3⁺ bEVs upon the adoptive transfer of WT mouse splenocytes into _Rag1⁻/⁻_ mice. Right, quantification graphs. n = 3 (_Rag1⁻/⁻_, no transfer), n = 4 (_Rag1⁻/⁻_, +WT splenocyte transfer), and n = 3 (WT, no transfer). **g** Waterfall plot representing the

bEV miRNAs whose expression significantly changed (adjusted _p_ < 0.05) by the adoptive transfer of WT T cells into _Rag1⁻/⁻_ mice. These miRNAs are regarded as T cell-dependent miRNAs. _Rag1⁻/⁻_ mice, n = 6; _Rag1⁻/⁻_ mice + WT T cells, n = 6. **h** T cell-dependent bEV-associated miRNAs (bEV-miRNAs) were over-represented by the miRNAs whose expression changes were previously observed in the mice with sociability deficits. The list of miRNAs were curated from previous publications (at least 2 publications for each mouse model) on several mouse models with sociability impairment[77–101]. Enrichment was calculated with Fisher's exact test. **i** Potential target genes of T cell-dependent EV miRNAs were enriched in the regulation of synapses and neurons. Each bar graph represents mean ± SEM. Each dot represents one mouse. ns, not significant. *_p_ < 0.05, ***_p_ < 0.005, ****_p_ < 0.001. Significance was determined by Student's _t_-test and one-way ANOVA with _post hoc_ Dunnett's test. See Supplementary Data 13 for details of the statistical analysis. Source data are provided as a Source Data file.

_Rag1⁻/⁻_ mice (Fig. 4d, e), suggesting that WT bEVs normalize mPFC excitability. These data indicate that WT bEVs increase sociability in _Rag1⁻/⁻_ mice, at least in part, by attenuating the mPFC hyperexcitability.

## WT bEVs restore inhibitory postsynaptic activities in the mPFC

We next used whole-cell patch clamp recordings in brain slices to assess the impact of WT bEVs on synaptic activities in the mPFC. _Rag1⁻/⁻_ mice exhibited significantly reduced amplitudes of spontaneous inhibitory postsynaptic currents (sIPSCs) compared to WT mice (Fig. 5a, b), while sIPSC frequencies showed a trending decrease. AMPA receptor-mediated spontaneous excitatory postsynaptic currents were unchanged between WT and _Rag1⁻/⁻_ mice (Supplementary Fig. 9a, b). Notably, incubation of _Rag1⁻/⁻_ brain slices with WT bEVs restored sIPSC amplitudes to WT levels (Fig. 5a, b). These findings suggest that impaired inhibitory postsynaptic signaling contributes to the mPFC hyperexcitability in _Rag1⁻/⁻_ mice and is targeted by WT bEV. A similar beneficial effect was observed in _Cntnap2⁻/⁻_ mice (Fig. 5c, d), indicating that modulation of mPFC inhibitory postsynaptic signaling by WT bEV may be a shared mechanism underlying sociability improvement.

## PFC transcriptomes change in _Rag1⁻/⁻_ mice and by WT bEV administration

To gain insight into the molecular mechanisms underlying EV-mediated modulation of mPFC neuronal activities, we performed RNA-seq analysis of the PFC of _Rag1⁻/⁻_ mice that received intravenous injection of WT bEVs. We also compared the PFC gene expression profiles between WT and _Rag1⁻/⁻_ mice. WT bEV injection altered the expression of 2027 genes in the _Rag1⁻/⁻_ PFC (Fig. 5e and Supplementary Data 5–8). Of these, 1280 genes overlapped with the 4772 genes differentially expressed between WT and _Rag1⁻/⁻_ PFC (Fig. 5e and Supplementary Data 5–8). Pathway enrichment analysis revealed that these genes were enriched for synaptic regulation (Fig. 5f and Supplementary Data 9), indicating that WT bEVs modulate synapse-related gene expression in the _Rag1⁻/⁻_ mPFC.

## Small RNAs mediate sociability rescue by WT bEVs in _Rag1⁻/⁻_ mice

EV-associated miRNAs (EV miRNAs) have been shown to influence gene expression in recipient cells, even in a distant organ[14,15,18]. To test whether miRNAs mediate the behavioral and synaptic effects of WT bEVs, we first extracted small RNA fractions from WT bEVs and loaded them into _Rag1⁻/⁻_ bEVs, as previously described[41]. These reconstituted bEVs (_Rag1⁻/⁻_ bEVs + WT RNA) were intravenously injected into _Rag1⁻/⁻_ mice. Sociability was then compared with _Rag1⁻/⁻_ mice receiving _Rag1⁻/⁻_ bEVs + _Rag1⁻/⁻_ RNA and WT bEVs + WT RNA. _Rag1⁻/⁻_ mice receiving _Rag1⁻/⁻_ bEVs + WT RNA showed significantly enhanced sociability, comparable to those receiving WT bEVs + WT RNA and higher than those receiving _Rag1⁻/⁻_ bEVs + _Rag1⁻/⁻_ RNA (Supplementary Fig. 10a, b).

These findings demonstrate that small RNAs associated with WT bEVs, most likely miRNAs, are sufficient to rescue sociability in _Rag1⁻/⁻_ mice.

## _Rag1⁻/⁻_ mice show altered expression of miR-23a-3p and miR-103-3p in bEVs and their predicted target, PKCε, in the mPFC

To identify the miRNA cargo and their target mRNA by which WT bEVs increase sociability and modulate mPFC neuronal activities, we further analyzed our EV miRNA-seq (Fig. 1g and Supplementary Data 1) and PFC RNA-seq datasets (Fig. 5e, f and Supplementary Data 5-8). We focused on 755 mRNAs that were upregulated in _Rag1⁻/⁻_ PFC compared to WT and downregulated by WT bEV injection (Fig. 6a and Supplementary Data 10). These mRNAs were potential targets of WT bEV miRNAs that were reduced in _Rag1⁻/⁻_ mice. As WT bEVs enhanced inhibitory postsynaptic activities in the mPFC of _Rag1⁻/⁻_ mice (Fig. 5a, b), we prioritized mRNAs implicated in GABAergic synapse regulation and identified seven candidates (Fig. 6a). Comparing these mRNAs with predicted targets of the 40 miRNAs whose expression increased by WT T cell transfer in _Rag1⁻/⁻_ bEVs (Fig. 1g and Supplementary Data 11, 12), we identified _Prkce_, encoding protein kinase C epsilon (PKCε), as a plausible target. PKCε has been shown to reduce GABA-mediated postsynaptic inhibitory currents by downregulating postsynaptic GABA_A receptor localization[42]. We indeed found increased PKCε protein levels in the _Rag1⁻/⁻_ mPFC neurons by immunohistochemistry (Fig. 6b, c). Consistent with a previous study reporting that PKCε downregulates synaptic GABA_A receptor levels[42], we observed that synaptic GABA_A receptor localization [GABA_A receptor subunit (GABA_ARγ2) associated with Gephyrin] on CaMKIIα⁺ neurons decreased in the _Rag1⁻/⁻_ mPFC (Fig. 6d, e). We also observed that two miRNAs predicted to target PKCε, miR-23a-3p and miR-103-3p (Supplementary Fig. 11a), were present in WT bEVs but significantly reduced in _Rag1⁻/⁻_ bEVs (Fig. 6f). Notably, miR-23a-3p was also downregulated in bEVs from _Shank3⁻/⁻_ and _Cntnap2⁻/⁻_ mice (Supplementary Fig. 11b). These findings suggest that WT bEV-associated miRNAs, particularly miR-23a-3p and miR-103-3p, modulate mPFC inhibitory postsynaptic signaling by regulating PKCε expression and synaptic GABA_A receptor localization.

## Intravenous injection of _Rag1⁻/⁻_ bEVs loaded with miR-23a-3p and miR-103-3p increases sociability and reduces mPFC hyperexcitability

We next tested whether miR-23a-3p and miR-103-3p could impact the mPFC neuronal alterations and increase sociability in _Rag1⁻/⁻_ mice. We loaded _Rag1⁻/⁻_ bEVs with miRNA mimics of miR-23a-3p and miR-103-3p and intravenously injected them into _Rag1⁻/⁻_ mice (Fig. 7a). _Rag1⁻/⁻_ mice receiving these reconstituted bEVs significantly improved sociability compared to those receiving _Rag1⁻/⁻_ bEVs loaded with non-targeting control mimics (Fig. 7b). The behavioral improvement was accompanied by reduced expression of neuronal c-Fos and PKCε, and increased levels of synaptic GABA_A receptor in the mPFC (Fig. 7c–h). Finally, brain slice electrophysiology revealed that _Rag1⁻/⁻_

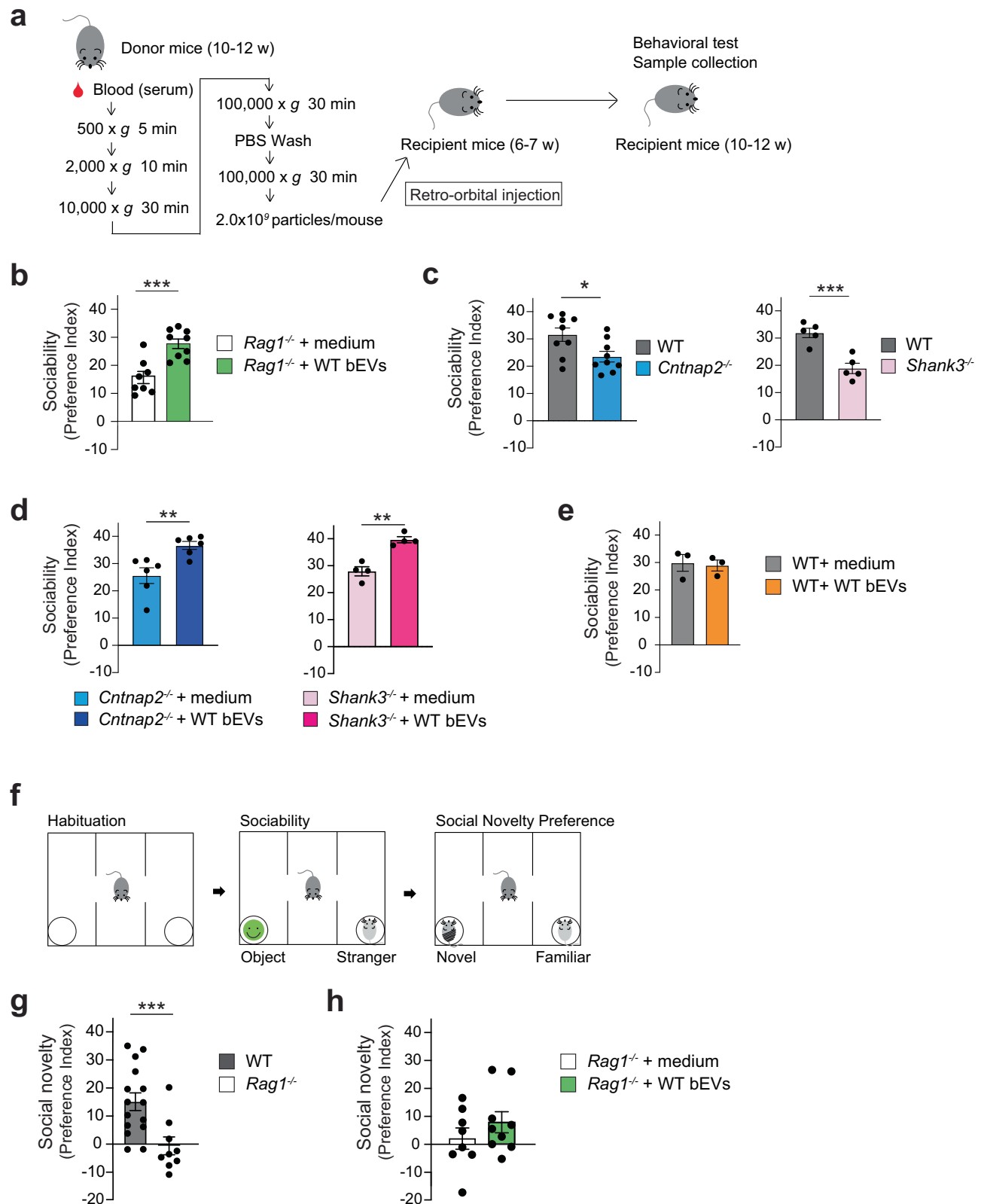

bEVs loaded with miR-23a-3p and miR-103-3p significantly enhanced sIPSC amplitudes in the mPFC pyramidal neurons compared to $Rag1^{-/-}$ bEVs loaded with non-targeting control mimics (Fig. 7i, j). These data indicate that miR-23a-3p and miR-103-3p are key mediators of bEV-induced improvement of sociability and mPFC inhibitory neurotransmission through regulation of PKCε and synaptic GABA$_A$ receptor (Fig. 7k).

## miR-23a-3p in T cell-derived EVs contributes to sociability

We finally examined the source of pro-social EV miRNAs, miR-23a-3p and miR-103-3p. First, we excluded the possibility that these miRNAs were merely co-precipitated with EVs during bEV collection. As EV miRNAs are protected from RNase degradation unless the vesicle membrane is disrupted (e.g., by TrixonX-100 treatment)[43,44], we treated bEVs with RNase in the presence or absence of 1% Triton-X100.

**Fig. 2 | WT bEVs improve sociability in *Rag1*$^{-/-}$ mice and two additional mouse models with sociability deficits. a** Experimental outline of bEV collection, intravenous injection, and subsequent behavioral assay. **b** Intravenous injection of WT bEVs into *Rag1*$^{-/-}$ mice increased their sociability. *Rag1*$^{-/-}$ mice + medium, $n = 8$; *Rag1*$^{-/-}$ mice + WT EVs, $n = 9$. **c** *Cntnap2*$^{-/-}$ and *Shank3*$^{-/-}$ mice showed a reduced preference for interacting with the mouse versus the object (sociability deficits) in the three-chamber social interaction test. Left, WT mice ($n = 9$) and *Cntnap2*$^{-/-}$ mice ($n = 9$). Right, WT mice ($n = 5$) and *Shank3*$^{-/-}$ mice ($n = 5$). **d** Intravenous injection of WT bEVs into *Cntnap2*$^{-/-}$ and *Shank3*$^{-/-}$ mice increased their sociability. Left, *Cntnap2*$^{-/-}$ mice injected with medium or WT bEVs ($n = 6$ per group). Right, *Shank3*$^{-/-}$ mice injected with medium or WT bEVs ($n = 4$ per group). **e** Intravenous injection of WT bEVs did not enhance sociability in WT mice. WT mice + medium, $n = 3$; and WT mice + WT bEVs, $n = 3$. **f** Outline of the three-chamber social novelty preference test. **g** *Rag1*$^{-/-}$ mice also showed a reduced preference for interacting with a novel mouse versus a familiar mouse (social novel preference deficits) in the three-chamber social interaction test. WT mice, $n = 15$; and *Rag1*$^{-/-}$ mice, $n = 9$. **h** Intravenous injection of WT bEVs did not increase social novelty preference in *Rag1*$^{-/-}$ mice. *Rag1*$^{-/-}$ mice + medium, $n = 8$; *Rag1*$^{-/-}$ mice + WT EVs, $n = 9$. Each bar graph represents mean ± SEM. Each dot represents one mouse. ns, not significant. *$p < 0.05$, **$p < 0.01$, ***$p < 0.005$. Significance was determined by Student's *t*-test. See Supplementary Data 13 for details of the statistical analysis. Source data are provided as a Source Data file.

Levels of miR-23a-3p and miR-103-3p were reduced only in the presence of 1% TritonX-100 (Supplementary Fig. 12a, b), indicating that these miRNAs are primarily located inside the EVs, although a fraction may be on the surface.

We then investigated the cellular origin of these miRNAs. Since T cell transfer into *Rag1*$^{-/-}$ mice increased their levels in bEVs (Fig. 1f), we tested whether T cells secrete EVs containing these miRNAs. T cells were purified from spleens, cultured, and EVs were collected from the conditioned medium (Fig. 8a). Notably, miR-23a-3p and, at a lesser level, miR-103-3p were detected in T cell-derived EVs (Fig. 8b and Supplementary Fig. 13a). Following intravenous injection into *Rag1*$^{-/-}$ mice, these cultured T cell-derived EVs were detected in the mPFC and co-localized with neurons and microglia (Fig. 8c, d, and Supplementary Fig. 13b), similar to the whole bEVs (Fig. 3) and in vivo generated T cell-derived EVs (Supplementary Fig. 13c–f), supporting their physiological relevance.

We finally determined whether T cell-derived miR-23a-3p contributed to the effects of WT bEVs on sociability using *Mir23a*$^{-/-}$ mice[45]. *Mir23a*$^{-/-}$ bEVs failed to restore sIPSC amplitudes in the PFC of *Rag1*$^{-/-}$ and *Cntnap2*$^{-/-}$ mice (Supplementary Fig. 14a, b), consistent with the role of miR-23a-3p in mPFC inhibitory neurotransmission (Fig. 7i, j). We purified T cells from *Mir23a*$^{-/-}$ mice and compared their ability to influence sociability deficits in *Rag1*$^{-/-}$ mice with that of WT T cells (Fig. 8e). Despite successful T cell reconstitution (Fig. 8f, g), *Rag1*$^{-/-}$ mice receiving *Mir23a*$^{-/-}$ T cells exhibited reduced levels of miR-23a-3p in bEVs (Fig. 8h), indicating that T cells are a major source of this miRNA in circulating EVs. Indeed, adoptive transfer of *Mir23a*$^{-/-}$ T cells failed to increase sociability in *Rag1*$^{-/-}$ mice (Fig. 8i). Together, these findings demonstrate that miR-23a-3p is secreted in EVs from T cells and contributes to sociability.

## Discussion

This study reveals that T cell-derived EV miRNAs in circulating EVs regulate sociability in mice. We demonstrated that circulating EVs enter the brain parenchyma and co-localize with neurons and microglia in the mPFC. At the molecular level, EV miRNAs, particularly miR-23a-3p and miR-103-3p, reduce PKCε expression, enhance synaptic GABA$_A$ receptor localization, and increase inhibitory postsynaptic signaling in the *Rag1*$^{-/-}$ mPFC pyramidal neurons. We further identified miR-23a-3p and miR-103-3p in T cell-derived EVs that co-localize with neurons in the mPFC. Notably, *Mir23a*$^{-/-}$ T cells failed to increase sociability in *Rag1*$^{-/-}$ mice and *Mir23a*$^{-/-}$ bEVs did not enhance sIPSC in *Rag1*$^{-/-}$ mPFC pyramidal neurons. Beneficial effects of WT bEVs were also observed in *Cntnap2*$^{-/-}$ and *Shank3*$^{-/-}$ mice, and *Mir23a*$^{-/-}$ bEVs were unable to increase sIPSC in *Cntnap2*$^{-/-}$ mice, implicating miR-23a-3p loss as a shared mechanism underlying inhibitory synaptic dysfunction in the mPFC across models of sociability deficits. Therefore, our findings provide conceptual and mechanistic insights into how circulating T cell-derived EV miRNAs influence brain synaptic activity and sociability, and suggest that EVs and miRNAs may have therapeutic potential for sociability impairment.

PKCε has been shown to downregulate synaptic GABA$_A$ receptors via the N-ethylmaleimide-sensitive factor and reduce inhibitory postsynaptic currents in the hippocampus[42]. Our study shows that PKCε-dependent modulation of GABAergic signaling is also essential in the mPFC, where its dysregulation leads to neuronal hyperexcitability. This modulation is further regulated by circulating EV-associated miRNAs, miR-23a-3p and miR-103-3p. Considering that blood EVs are also detected in the hippocampus and cerebellum, similar EV miRNA-mediated modulation of synaptic neurotransmission may occur in other brain regions. Previous studies have reported that local protein translation controls the expression of GABA$_A$ receptors and some PKC isoforms[46–48]; thus, EV miRNAs may directly influence the local translation of PKCε at synapses, regulating synaptic GABA$_A$ receptor level. It remains unclear whether and how the expression changes in a pleiotropic enzyme, PKCε, preferentially influence GABAergic signaling in the mPFC. As the current knowledge of predicted targets by miRNAs is still limited, we cannot completely exclude that miR-23a-3p and miR-103-3p may regulate GABAergic signaling via molecules other than PKCε.

While earlier studies have focused on the role of cytokines, canonical effector molecules of T cells, in regulating neuronal activity and behavior[4,5,7,49], our data uncover an additional miRNA-mediated layer of T cell–neuron communication. Specifically, we identify miR-23a-3p and miR-103-3p as pro-social EV miRNAs derived from T cells. T cell-derived EVs and their miRNAs are known to modulate immune cell function[50,51], but their role outside the context of immune responses has been largely unexplored. Our findings establish a paradigm that miRNAs mediate immune cell communication with neurons in a more physiological context. Our study also defines T cells as an important source of circulating EVs and miRNAs that mediate the blood-brain communication, regulating neuronal function and behavior. Future studies will determine which T cell subsets (e.g., CD4$^+$ T cells, γδ T cells) contribute to pro-social EV miRNA production. Currently, the lack of cell-type-specific EV markers limits our ability to trace specific EVs in circulation back to their cellular origin, but such challenges will be overcome soon with rapid advances in EV research.

The EV uptake by brain cells was selective. We observed substantial overlaps of bEV signals with neurons and microglia, but not astrocytes or oligodendrocytes. As macrophages are known to take up EVs in the periphery and circulation[52,53], microglial uptake of bEVs may not be surprising and is likely to be related to their homeostatic function. In contrast, neuronal uptake of bEVs is unexpected because neurons are not professional phagocytes. Although further research is required, bEV uptake by neurons may depend on some specific mechanisms (e.g., receptors for certain EV surface proteins) and be selective for certain types of EVs (e.g., EVs with surface protein modifications).

Notably, the behavioral improvement was also selective. WT bEVs improved sociability but not social novelty preference in *Rag1*$^{-/-}$ mice. Similarly, WT serum, splenocytes, and T cells failed to improve social novelty preference. Thus, circulating T cell-derived EVs may preferentially target sociability-related neuronal circuitry, highlighting the significance of possible EV tropism for specific cell types or brain regions. One limitation of this study is the fact that the *Rag1*$^{-/-}$ mouse sociability scores were variable across different experiments, which

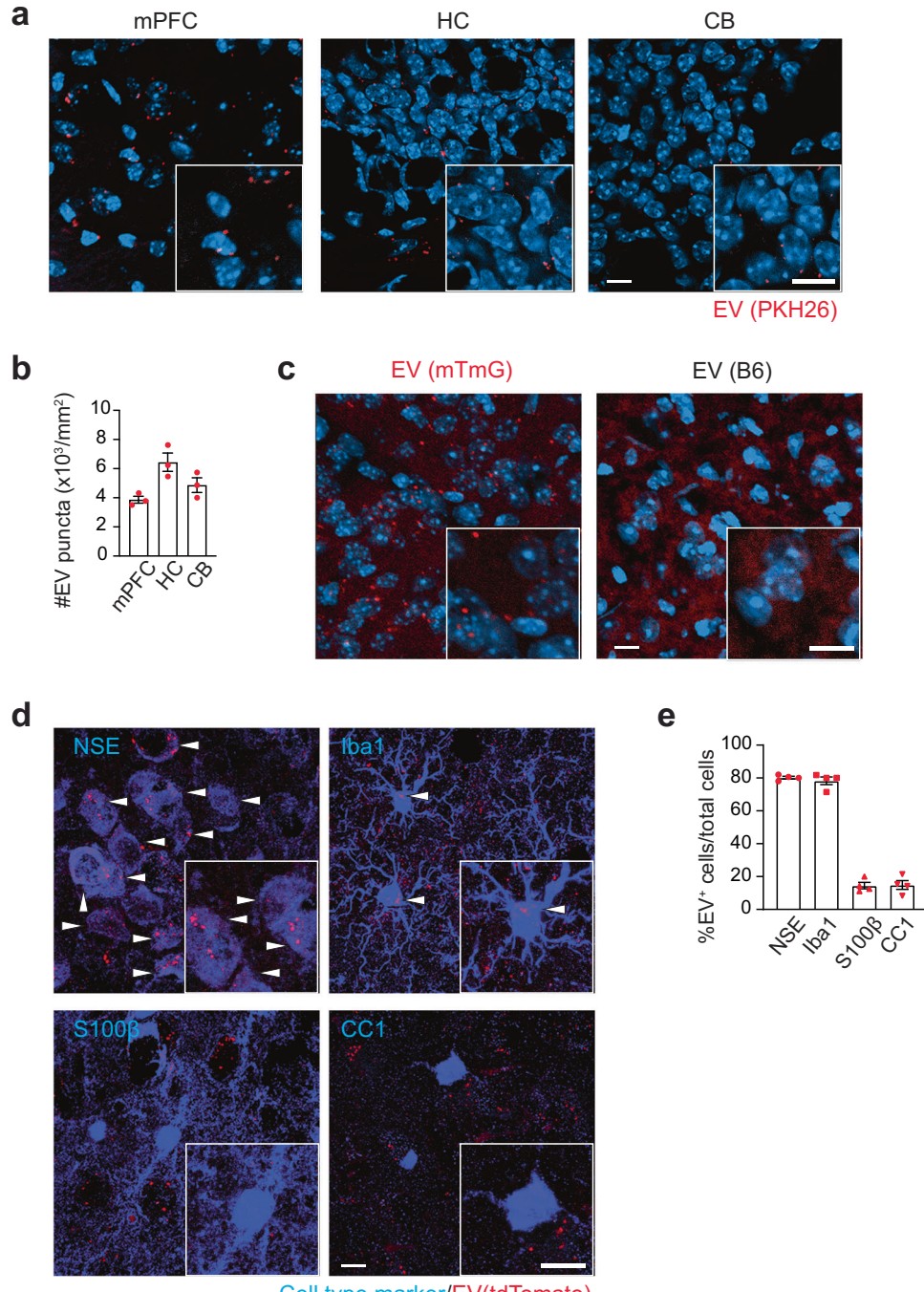

**Fig. 3 | Intravenously injected WT bEVs are co-localized to neurons and microglia in the brain parenchyma. a** Representative confocal microscope images showing the distribution patterns of intravenously injected PKH26-labeled EVs (red) in the mPFC, hippocampus (HC), and cerebellum (CB) of $Rag1^{-/-}$ mice. DAPI stain (blue). **b** Quantification of PKH26- labeled EVs in the brains of $Rag1^{-/-}$ mice ($n = 3$). **c** Representative confocal microscope images showing the distribution patterns of intravenously injected tdTomato⁺ EVs (red) from mTmG mice in the mPFC of $Rag1^{-/-}$ mice. DAPI stain (blue). **d** Representative confocal microscope images showing the localization of tdTomato⁺ EVs with neurons (NSE⁺ cells) and microglia (Iba1⁺ cells) in the mPFC of $Rag1^{-/-}$ mice. EV⁺ cells are indicated by white arrowheads. **e** Quantification of tdTomato⁺ EVs colocalized with neurons (NSE⁺ cells), microglia (Iba1⁺ cells), astrocytes (S100β⁺ cells), and oligodendrocytes (CC1⁺ cells) in the mPFC of $Rag1^{-/-}$ mice ($n = 4$). Scale bars, 10 μm. Each bar graph represents mean ± SEM. Each dot represents one mouse. Source data are provided as a Source Data file.

was likely caused by multiple factors, particularly the changes in mouse housing conditions and behavioral rooms that we experienced during this study. However, our data consistently showed the improvement of sociability by WT bEVs, underscoring the beneficial effects of WT bEVs.

We observed the beneficial effects of bEVs on sociability in $Cntnap2^{-/-}$ and $Shank3^{-/-}$ mice as well. Nevertheless, the underlying mechanisms have not been thoroughly dissected in the current study. Although we showed that bEVs improved electrophysiological phenotypes in a miR-23a-dependent manner in $Cntnap2^{-/-}$ mice, it remains unclear whether the PKCε-GABA$_A$ receptor axis is similarly impaired in these mice. It is also unclear whether the beneficial effects of bEVs on $Shank3^{-/-}$ mice are dependent on miR-23a-3p. Further studies are required to determine the common and selective mechanisms

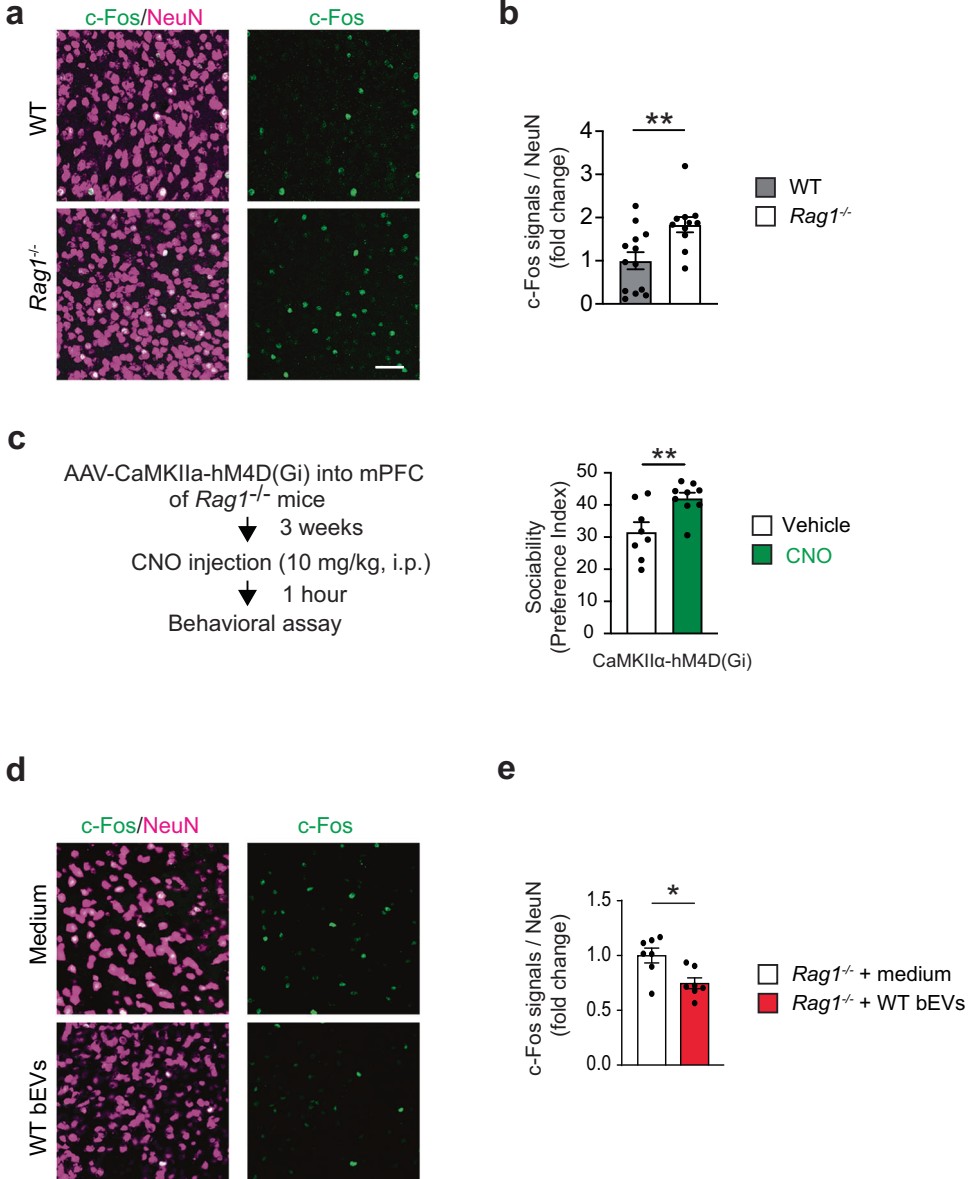

**Fig. 4 | WT bEVs attenuate hyperexcitability in the mPFC of _Rag1_⁻/⁻ mice.**
**a** Representative confocal microscope images showing enhanced c-Fos immunoreactivity in the mPFC of _Rag1_⁻/⁻ mice. **b** Quantification of c-Fos-positive neurons as fold change of percentages for c-Fos⁺ cells among NeuN⁺ cells. WT mice, _n_ = 13; _Rag1_⁻/⁻ mice, _n_ = 11. **c** Chemogenetic inhibition of CaMKIIα⁺ neuronal activities in the mPFC by hM4D(Gi) upon CNO injection increased sociability in _Rag1_⁻/⁻ mice. Vehicle _n_ = 8 mice; CNO _n_ = 9 mice. **d** Representative confocal microscope images showed a reduction of c-Fos immunoreactivity in the mPFC of _Rag1_⁻/⁻ mice after WT bEV injection. **e** Quantification of c-Fos-positive neurons as fold-change of percentages for c-Fos⁺ cells among NeuN⁺ cells. _Rag1_⁻/⁻ mice + medium, _n_ = 7; _Rag1_⁻/⁻ mice + WT EVs, _n_ = 7. Scale bar, 50 µm. Each bar graph represents mean ± SEM. Each dot represents one mouse. ns, not significant. *_p_ < 0.05, **_p_ < 0.01, ****_p_ < 0.001. Significance was determined by Student's _t_-test. See Supplementary Data 13 for details of the statistical analysis. Source data are provided as a Source Data file.

underlying the effects of WT bEVs on sociability across multiple mouse models.

Our findings are also consistent with previous human studies. Altered expression of miR-23a-3p and miR-103-3p has been reported in the brain tissues and blood of patients with autism spectrum disorders (ASD) and schizophrenia[54–57], conditions accompanied by sociability deficits. miR-23a-3p is decreased in the peripheral blood of patients with schizophrenia[55], while miR-103-3p is consistently decreased across multiple studies with ASD patients[56,57]. Notably, one study also found that the miR-23a-3p targets genes related to synaptic function and ASD risks[54]. Nevertheless, the behavioral consequences of these miRNA changes remain unclear. Future studies should investigate whether the blood levels of miR-23a-3p and miR-103-3p correlate with sociability and other cognitive functions in these patients and may

have therapeutic potentials as observed in multiple mouse models with sociability deficits in this study.

Finally, our work underscores the need to investigate blood EV changes not only as biomarkers but also as possible pathological mechanisms in psychiatric and neurological disorders[14,58]. While blood EV alterations are recognized as useful biomarkers, our research indicates that they may also contribute to disease mechanisms involving periphery-brain communication. Importantly, our work demonstrated that the supplementation of EVs and miRNAs improved sociability. Although engineering cell culture-derived EVs is currently a popular therapeutic approach, our findings suggest that modulating autologous circulating EVs and miRNAs may be another promising strategy to restore neuronal function and behaviors under pathological conditions.

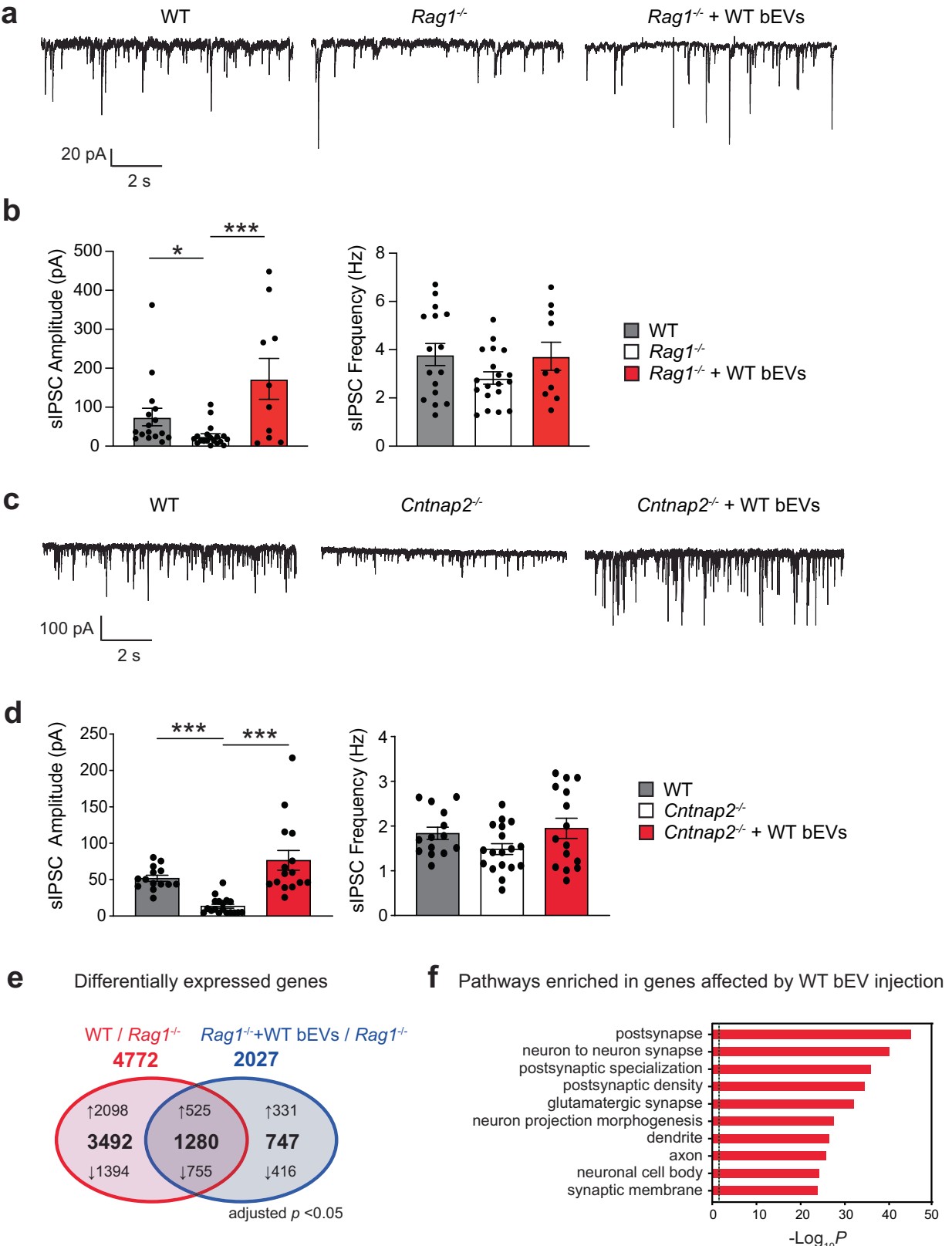

**a**  WT          *Rag1⁻/⁻*          *Rag1⁻/⁻* + WT bEVs

20 pA | 2 s

**b**

**c**  WT          *Cntnap2⁻/⁻*          *Cntnap2⁻/⁻* + WT bEVs

100 pA | 2 s

**d**

**e** Differentially expressed genes

**f** Pathways enriched in genes affected by WT bEV injection

## Methods

### Mice

*Rag1⁻/⁻*, *Cntnap2⁻/⁻*, and *Shank3⁻/⁻*, mTmG, and C57BL/6 mice (Stock# 002216, 017482, 017688, 007676, and 000664) were purchased from the Jackson Laboratory and housed in specific pathogen-free facilities at the Johns Hopkins University and the University of Alabama at Birmingham. *Mir23a⁻/⁻* mice were obtained from Dr. Richard Dahl (Indiana University, USA). All the mice were on C57BL/6 background. Male mice were used for the experiments at 8–12 weeks of age unless stated otherwise. All experimental procedures were performed under the animal protocols approved by the Johns Hopkins University and University of Alabama at Birmingham

**Fig. 5 | WT bEVs restore impaired postsynaptic GABAergic currents in mPFC pyramidal neurons of *Rag1*⁻ᐟ⁻ mice. a** Representative traces of spontaneous inhibitory postsynaptic currents (sIPSCs) in the mPFC (PrL) pyramidal neurons for WT, *Rag1*⁻ᐟ⁻, and *Rag1*⁻ᐟ⁻ slices exposed to WT bEVs. Voltage was clamped at −70 mV, and recordings were taken for 5 min. AP-V (50 μM) and DNQX (50 μM) were added to the bath to inhibit NMDA and AMPA receptors throughout the experiment. The temperature was set at 30 ± 1 °C. **b** Quantification of sIPSC amplitude and frequency among groups. 10–19 cells from three different mice were used for each group. Each bar graph represents mean ± SEM. Each dot represents one cell. **c** Representative traces of sIPSCs in the mPFC pyramidal neurons for WT, *Cntnap2*⁻ᐟ⁻, and *Cntnap2*⁻ᐟ⁻ slices exposed to WT bEVs. **d** Quantification of sIPSC amplitude and frequency between groups. 14–18 cells from three different mice were used for each group. Each bar graph represents mean ± SEM. Each dot represents one cell. **e** Venn diagram showing the numbers of differentially expressed genes in the mPFC of *Rag1*⁻ᐟ⁻ mice compared to WT mice and those in the mPFC of *Rag1*⁻ᐟ⁻ mice with WT bEV injection compared to *Rag1*⁻ᐟ⁻ mice with medium injection. WT mice, *n* = 3; *Rag1*⁻ᐟ⁻ mice, *n* = 3; *Rag1*⁻ᐟ⁻ mice + WT EVs, *n* = 3. **f**, Pathway enrichment analysis of 1280 genes whose expression were altered in opposite directions between WT vs *Rag1*⁻ᐟ⁻ and *Rag1*⁻ᐟ⁻ vs *Rag1*⁻ᐟ⁻+ WT bEVs comparisons. $*p < 0.05$, $***p < 0.005$. Significance was determined by one-way ANOVA and Kruskal–Wallis tests with respective multiple comparison tests. See Supplementary Data 13 for details of the statistical analysis. Source data are provided as a Source Data file.

Institutional Animal Care and Use Committees (#MO17M204, #21539, respectively).

### Euthanasia and anesthesia method

During euthanasia, enough care was taken to minimize the suffering of the mice. For non-survival procedures, anesthesia was induced with 5% isoflurane, and maintained until spontaneous respiration completely ceased. A secondary physical method was used to confirm death. For survival procedures, anesthesia was induced with 5% isoflurane and maintained at 1–2% during surgery. For analgesia, 5 mg/kg Carprofen or 0.1 mg/kg Buprenorphine Hydrochloride was administered intraperitoneally before surgery and every 12 h for 24 h postoperatively.

### Behavioral assays

Behavioral assays were performed on male mice at 8–12 weeks of age. All the assays were conducted between 10 am and 3 pm during the light phase. Independent cohorts of mice were tested in the three-chamber social interaction test, the open field test, and the buried food pellet test.

**Three-chamber social interaction test.** The three-chamber social interaction test was conducted as previously described[59,60]. All mice were tested in a nonautomated three-chambered box. Dividing walls had retractable doorways allowing access into each chamber. Mice were acclimated to the three-chambered box for 4 days before the test (10 min/day). On the test day, mice were transported to the testing room and habituated for at least 1 h before the experiment. A white noise generator was used to mitigate any unforeseen noises. The subject mouse was habituated in the chamber with two empty cylinders for 10 min. Then, the "toy" object was placed in one of the cylinders, and mouse (stranger 1) was placed in another cylinder for the "sociability" trial. Mice were allowed to explore the chambers for 10 min. In the next "social novelty preference" trial, stranger 1 was kept in the cylinder as the familiar mouse, and the toy object was replaced with a novel mouse (stranger 2). The subject mouse was again allowed to explore the chambers. During these two trials, mouse activities were recorded on video, and the time spent sniffing each cylinder was manually measured. Preference index was calculated as follows. For sociability test: (sniffing time to mouse) × 100 / (sniffing time to mouse + sniffing time to object) - 50. For social novelty preference test: (sniffing time to novel mouse) × 100 / (sniffing time to novel mouse + sniffing time to familiar mouse) - 50.

**Buried food pellet test.** The buried food pellet test was performed as previously described[61]. Mice were food-deprived for 24 h with free access to water. At the test, single mouse was placed in the test cage and allowed to freely explore the food pellets buried -0.5 cm below the surface of a 3 cm-deep layer of beddings. Latency to find the hidden food pellet was measured (*n* = 11–18 mice per group).

**Open field test.** Novelty-induced activity in the open field was assessed as described previously[60,62]. Locomotion, rearing, and center time were measured for 10 min using a Photobeam Activity System (PAS–Open Field, San Diego Instruments). The PAS system consisted of two vertically stacked frames, each containing infrared lasers arranged in a 16 × 16 grid, which detected mouse movement, including ambulation and rearing. The open field box and surrounding photobeam apparatus were housed in a ventilated cabinet. Single-beam breaks were automatically recorded as "counts" and the PAS system automatically started recording counts once the mouse started moving. The total counts were recorded and the percentages of center counts (defined as those in the central 27.5 × 27.5 cm area) to total counts were calculated.

### Serum and EV preparation

Blood (600–700 μl) was collected from mice via cardiac puncture under a deep anesthesia at time of sacrifice and placed at room temperature for 30 min. Sera were then separated by centrifugation at 300 × *g* for 5 min and stored at −80 °C for biochemical/molecular experiments or freshly used for in vivo experiments. For EV preparation, the sera were further centrifuged with 2000 × *g* for 10 min to remove platelets. EVs were prepared primarily with differential ultracentrifugation (UC), except for Supplementary Figs. 3 and 5 where precipitation (PPT) and size exclusion chromatography (SEC) were also used as indicated. For EV enrichment with differential ultracentrifugation, the sera were centrifuged at 10,000 × *g* for 30 min at 4 °C to remove large vesicles. Then, the sera were diluted with PBS at 1:4 ratio and ultracentrifuged at 100,000 × *g* for 30 min at 4 °C using Optima MAX-XP ultracentrifuge with a TLA 120.2 rotor (k factor = 11) (Beckman Coulter)[63]. The pellet was resuspended with 1 ml of PBS and ultracentrifuged at 100,000 × *g* for 30 min at 4 °C to wash, and then resuspended with 10 μL of PBS or DMEM medium. For EV precipitation, the sera were processed with the Total Exosome Isolation kit (Thermo Fisher Scientific) following the manufacturer's protocol, and the EV-containing fraction was resuspended in 50 μL PBS. For EV enrichment with size exclusion chromatography, each serum sample was applied to a 70 nm qEVsingle size exclusion column (Izon) and eluted with DPBS. Fractions were collected using an Izon automated fraction collector: following elution of a 1 ml void fraction, 0.5 ml fractions were collected, of which fractions 6 to 8 were enriched in EVs. These fractions were pooled, concentrated using Amicon 15 Ultra RC 100 kD filters, and aliquoted and stored in LoBind tubes (Eppendorf) at −80 °C.

### Transmission electron microscopy

Transmission electron microscopy (TEM) was performed using a negative-staining procedure at the University of Alabama at Birmingham Imaging Core Facility. Freshly prepared EV samples were adsorbed onto carbon-coated/palladium grids for 1 min to allow particle attachment. Excess liquid was blotted off, and the grids were negatively stained with 2% (w/v) uranyl acetate for 1 min following the facility's standard negative-staining protocol. After air-drying, EVs were visualized using a FEI Tecnai Spirit T12 transmission electron microscope (Thermo Fisher Scientific / FEI).

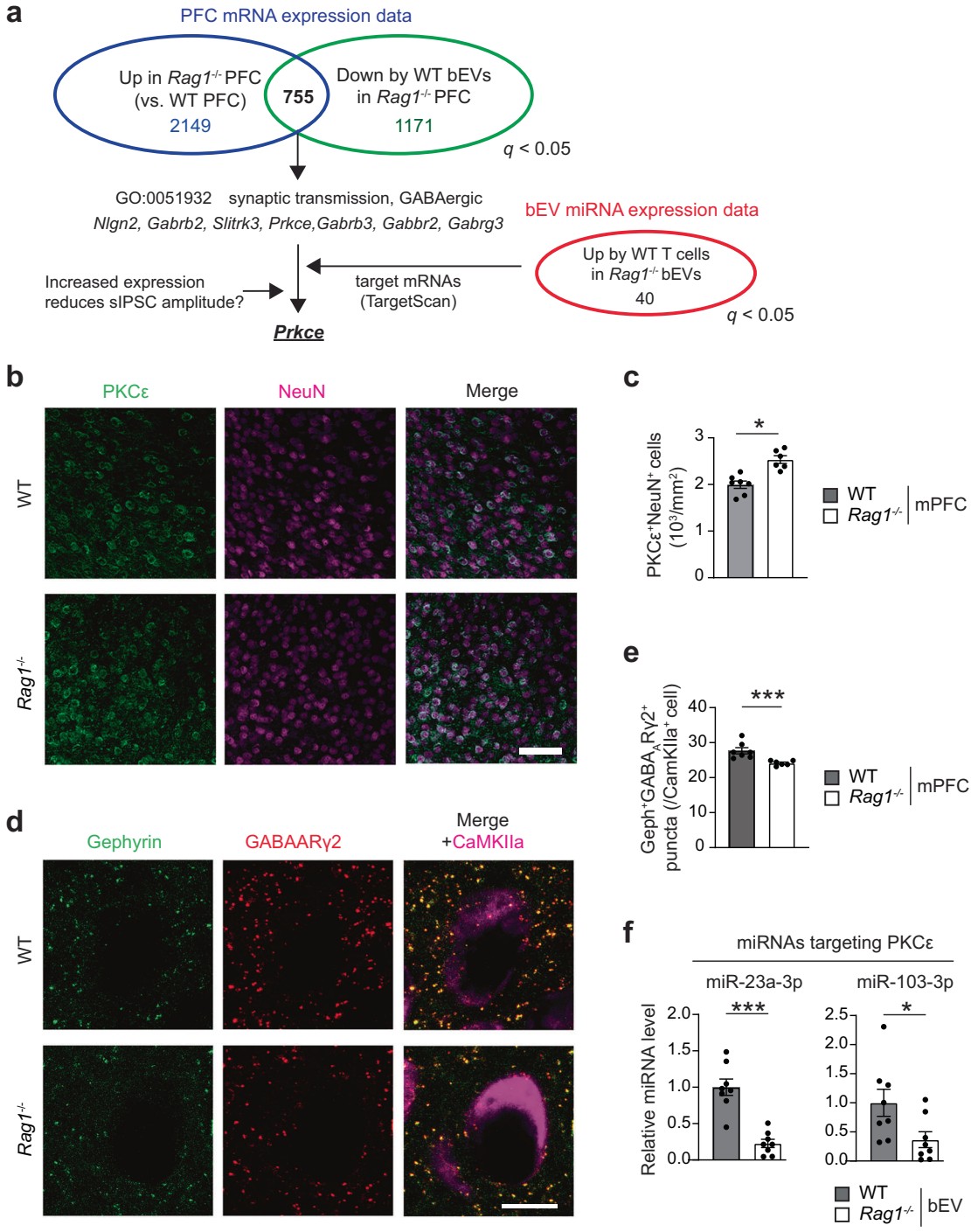

**Fig. 6 | *Rag1⁻/⁻* mice show increased expression of PKCε in the mPFC and a downregulation of its regulatory miRNAs, miR-23a-3p and miR-103-3p, in the blood EVs. a** Flow of data analysis to identify candidate genes responsible for GABAergic signaling changes in *Rag1⁻/⁻* mPFC. **b** Representative confocal microscope images showing enhanced neuronal PKCε immunoreactivity in the mPFC of *Rag1⁻/⁻* mice. **c** Quantification of PKCε⁺ neurons in the mPFC of WT and *Rag1⁻/⁻* mice. WT mice, *n* = 7 *Rag1⁻/⁻* mice, *n* = 6. **d** Representative confocal microscope images of Gephyrin and GABAₐRγ2 co-localization in CaMKIIα⁺ neurons of WT and *Rag1⁻/⁻*

mPFC. **e** Quantification of number of overlapped puncta of Gephyrin and GABAₐRγ2 in CaMKIIα⁺ neurons of WT and *Rag1⁻/⁻* mPFC. WT mice, *n* = 7; *Rag1⁻/⁻* mice, *n* = 6. **f** Expression levels of miR-23a-3p and miR-103-3p in the bEVs from WT and *Rag1⁻/⁻* mice (*n* = 8 per group). Data are shown as fold-change relative to WT data. Scale bar, 50 μm. Each bar graph represents mean ± SEM. Each dot represents one mouse. ns, not significant. *$p < 0.05$, ***$p < 0.005$. Significance was determined by Student's *t*-test. See Supplementary Data 13 for details of the statistical analysis. Source data are provided as a Source Data file.

## Nano particle tracking analysis (NTA)

Concentration and size distribution of particles in EV samples were analyzed with NanoSight NS300 (Malvern Panalytical) or ZetaView (Particle Metrix) following standard protocols[64,65]. For NanoSight analysis, the screen gain was consistently set to 1.0 and the camera level to 15. For calibration, the polyethylene beads (Malvern) diluted

1000-fold with Milli-Q water were used as a standard. After confirming that this concentration was $2-3 \times 10^8$/mL, the samples were measured.

## Adoptive cell transfer, serum injection, and EV injection

For adoptive cell transfer, splenocytes and T cells were collected from the spleen. T cells were isolated using the Pan T cell isolation

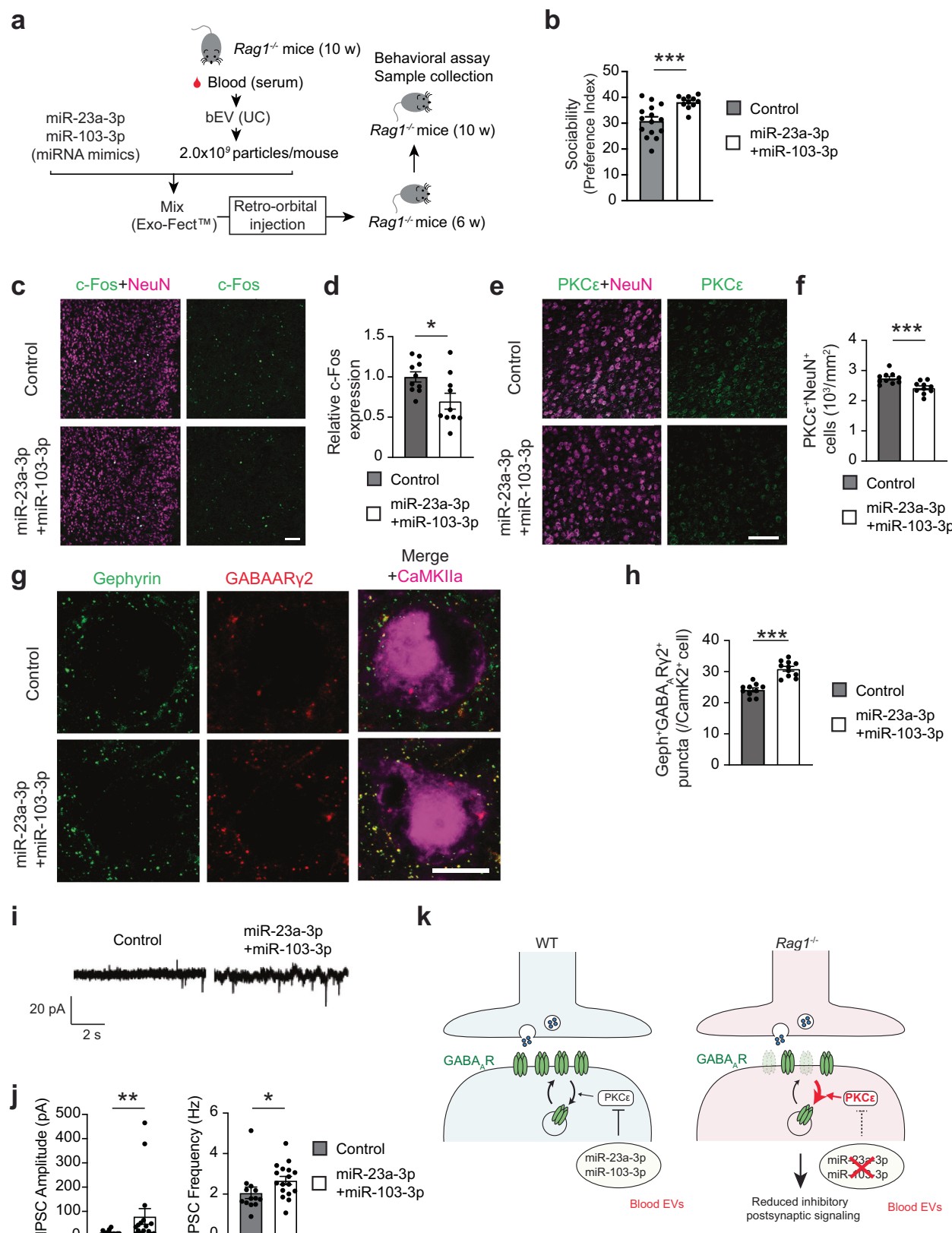

kit II, mouse (#130-095-130, Miltenyi Biotech). T cells were >99%-positive for CD3ε expression by flow cytometry. Splenocytes ($1 \times 10^8$ cells 100 µL per mouse) and T cells ($1 \times 10^7$ cells 100 µL per mouse) were transferred intravenously into *Rag1*[−/−] mice via retro-orbital injection following a standard protocol[66]. Sera (150 µL per mouse) and EVs ($2 \times 10^9$ particle in 100 µL per mouse unless stated

otherwise) were also transferred intravenously into WT and *Rag1*[−/−] mice via retro-orbital injection. In our method, the number of EVs recovered from 500 µL total venous blood was about $4 \times 10^9$ particles. Considering that the total blood volume of an adult mouse is 2 ml, our injection of $2 \times 10^9$ particles corresponds to about 12.5% volume of the total circulating EVs in the blood. This volume is less

**Fig. 7 | Supplementation of miR-23a-3p and miR-103-3p to *Rag1−/−* bEVs affects the mPFC molecular and functional changes in *Rag1−/−* mice and enhances their sociability. a** Experimental outline of bEV collection, miRNA mimics loading, intravenous injection, and subsequent behavioral assay. **b** Sociability data of *Rag1−/−* mice administered with *Rag1−/−* bEVs containing negative control miRNA mimics (control mimics) or miRNA mimics for miR-23a-3p and miR-103-3p (miR-23a-3p + miR-103-3p mimics). Control, *n* = 15; miR-23a-3p + miR-103-3p, *n* = 11. **c** Representative confocal microscope images of neuronal c-Fos expression in the mPFC of *Rag1−/−* mice administered with *Rag1−/−* bEVs containing control or miR-23a-3p + miR-103-3p mimics (*n* = 10 mice per group). **d** Quantification of c-Fos⁺ neurons in the mPFC (*n* = 10 per group). **e** Representative confocal microscope images of neuronal PKCε expression in the mPFC of *Rag1−/−* mice administered with *Rag1−/−* bEVs containing control or miR-23a-3p + miR-103-3p mimics. **f** Quantification of PKCε⁺ neurons in the mPFC (*n* = 10 per group). **g** Representative confocal microscope images of neuronal Gephyrin and GABA_ARγ2 (expression in CaMKIIα⁺ neurons in the mPFC of *Rag1−/−* mice administered with *Rag1−/−* bEVs containing control or miR-23a-3p + miR-103-3p mimics. **h** Quantification of number of Gephyrin and GABA_ARγ2 puncta in CaMKIIα⁺ neurons in the mPFC. Control, *n* = 10; miR-23a-3p + miR-103-3p, *n* = 11. *$p < 0.0001$. **i** Representative traces of sIPSCs of the mPFC neurons in brain slices from *Rag1−/−* mice administered with *Rag1−/−* bEVs containing control or miR-23a-3p + miR-103-3p mimics. **j** Quantification data of sIPSC amplitude and frequency changes by miR-23a-3p + miR-103-3p mimics in *Rag1−/−* mice (aggregated data from *n* = 3 mice per group). **k** Graphical summary of mechanism. Circulating EV miRNAs, including miR-23a-3p and miR-103-3p, inhibit the expression of PKCε in mPFC neurons, increasing synaptic GABA_A receptor localization and enhancing inhibitory postsynaptic signaling. Scale bars, 50 μm (**c**, **e**) and 10 μm (**i**). Each bar graph represents mean ± SEM. Each dot represents one mouse (**b**, **d**, **f**, **h**) or cell (**j**). ns, not significant. *$p < 0.05$, **$p < 0.01$, and ***$p < 0.005$. Significance was determined by Student's *t*-test. See Supplementary Data 13 for details of the statistical analysis. Source data are provided as a Source Data file.

than the maximal blood volume (=13%) we can collect per single blood donation in humans[67,68].

## Primary T cell culture

Mouse primary T cells were purified from spleens as Thy1.2⁺CD19⁻ cells by FACS sorting with a BD FACSAria II and cultured following a standard protocol[66]. Briefly, purified T cells ($1 \times 10^6$ cells/ml per well on 24-well plates) were stimulated with plate-bound anti-CD3ε (1 μg/ml) and anti-CD28 (1 μg/ml) and cultured for 36 and 60 h in RPMI complete medium (RPMI, 10% FBS, 1% non-essential amino acids, 1% sodium pyruvate, 0.05 mM B-Mercaptoethanol, and 1% Pen/Strep). Cell culture supernatants were collected, and EVs were collected by UC as described above.

## Western blot

EV and tissue lysates were prepared with RIPA buffer, separated on NuPAGE Bis-Tris Mini Gel (Thermo Fisher Scientific/Life Technologies) with equal amounts of total proteins loaded into each lane, and transferred to PVDF membrane (Millipore). After blocking in 5% skim milk/PBS-0.1% Tween® 20 (PBS-T) for 1 h, the membrane was incubated with the primary antibody overnight at 4 °C and then incubated with the HRP or fluorescent secondary antibodies for 1 h at room temperature. Chemiluminescence and fluorescence gel images were captured by ImageQuant LAS 4000 imager (GE Healthcare), Odyssey imaging system (Li-Cor Biosciences), and analyzed with ImageJ/Fiji software[4]. Images acquired using the Odyssey imaging system were converted to grayscale and inverted for Figures. The following primary antibodies were used: rabbit anti-CD9 (1:1000, #ab92726, Abcam, RRID: AB_10561589), rabbit anti-Alix (1:1000, #ab186429, Abcam, RRID: AB_2754981), rabbit anti-Calnexin (1:1000, #ADI-SPA-860-D, Enzo Life Science, RRID: AB_10616095), mouse anti-CD3ε (1:1000, #362701, BioLegend, RRID: AB_2563713), armenian hamster anti-CD3ε (1:1000, #100302, BioLegend, RRID: AB_312667), mouse anti-CD81 (1:1000, #sc-166029, Santa Cruz Biotechnology, RRID: AB_2275892), armenian hamster anti-mouse CD81(1:1000, #sc-18877, Santa Cruz Biotechnology, RRID: AB_627194), IRDye 680RD Donkey anti-mouse IgG(H+L) (1:10,000, LI-COR Biosciences #926-68072, RRID: AB_10953628), IRDye 800CW Donkey anti-rabbit IgG(H+L) (1:10,000, LI-COR Biosciences #926-32213, RRID: AB_621848), rabbit anti-Rag1 (1:1000, #sc-5599, Santa Cruz Biotechnology, RRID: AB_2300670), mouse anti-Rag1 (1:1000, #68591-1-Ig, Proteintech, RRID: AB_3085290), goat anti-Rag2 (1:1000, #sc-7623, Santa Cruz Biotechnology, RRID: AB_2175836), rabbit anti-TNFα (1:1000, #17590-1-AP, Proteintech, RRID: AB_2271853), and mouse anti-β-actin (1:5000, #sc-47778, Santa Cruz Biotechnology, RRID: AB_626632).

## Flow cytometry

Flow cytometry of splenocytes was performed using a standard protocol[66]. Single-cell suspensions were obtained from spleens collected after PBS perfusion. After lysis of red blood cell with 1× RBC Lysis Buffer (Thermo Fisher Scientific/eBioscience), cells were washed with FACS buffer and Fc-blocked with anti-CD16/CD32 (#14-0161-82, Thermo Fisher Scientific/eBioscience, RRID: AB_467133). Then, cells were stained with the following antibodies: rat anti-CD3 APC (1:100, #100235 BioLegend, RRID: AB_2561455), rat anti-mouse CD19 PE (1:100, #115508, BioLegend, RRID: AB_313643), rat anti- B220 Alexa Fluor 488 (1:100, #103228, BioLegend, RRID: AB_492874), rat anti-mouse CD3 Alexa Fluor 488 (1:100, #100210, BioLegend, RRID: AB_389301), anti-CD3 Alexa Fluor 700 (1:100, #100216, BioLegend, RRID: AB_493697), and rat anti-mouse CD45 FITC (1:100, #110705, BioLegend, RRID: AB_313494). Flow cytometry was conducted using FACS Calibur, LSRII, LSRFortessa, or Accuri C6 cytometer (all from BD Biosciences) and analyzed with FlowJo software (FlowJo LLC).

## DREADD experiments

Mice were stereotactically injected with 400 nl of pAAV-CaMKIIα-hM4D(Gi)-mCherry ($4.4 \times 10^{12}$ genomic copies/mL, #50477, Addgene) at the rate of 100 to 200 nl/min into the mPFC bilaterally using a NanoFil syringe with a 35G blunt needle (WPI). The following stereotactic coordinate was used for injection: anteroposterior (AP): +1.8 mm; mediolateral (ML): ±0.3 mm; and dorsoventral (DV): -2.1 mm from the bregma. Three to four weeks later, behavioral assays were conducted. CNO (10 mg/kg in 0.5% DMSO/PBS, #BML-NS105-0025, Enzo Life Sciences) was intraperitoneally injected 45 min prior to the three-chamber social interaction test. For the three-chamber social interaction test, mice were intraperitoneally injected with PBS during acclimation for 4 consecutive days and then injected with CNO or vehicle (0.5% DMSO/PBS) on the test day as previously described[69].

## Brain collection

Mice were anesthetized and transcardially perfused with ice-cold PBS. For RNA-seq experiments, brains were dissected and stored at −80 °C until use. For immunohistochemistry, mice were further perfused by 4% paraformaldehyde (PFA). Brains were dissected, post-fixed in ice-cold 4% PFA/PBS for 24 h, cryoprotected in 15% and 30% sucrose/PBS, and embedded in the O.C.T. compound.

## Immunohistochemistry

Immunohistochemistry experiments were conducted as previously described[60,70]. Free-floating coronal sections (40 μm in thickness) were prepared with a Leica cryostat and, if necessary, antigen retrieval was performed with 10 mM sodium citrate buffer (pH 8.5). The sections were then placed in blocking solution (PBS supplemented with 2% Normal Goat Serum, 1% BSA, 0.1% TritonX, 0.05% Tween-20, and 0.05% sodium azide) for 1 h at room temperature and then incubated at 4 °C overnight with the following primary antibodies: mouse anti-NeuN (1:500, #MAB377, Merck-Millipore, RRID: AB_2298772), rabbit anti-Iba1 (1:400, #019-19741, Wako Chemicals, RRID: AB_839504), goat anti-c-

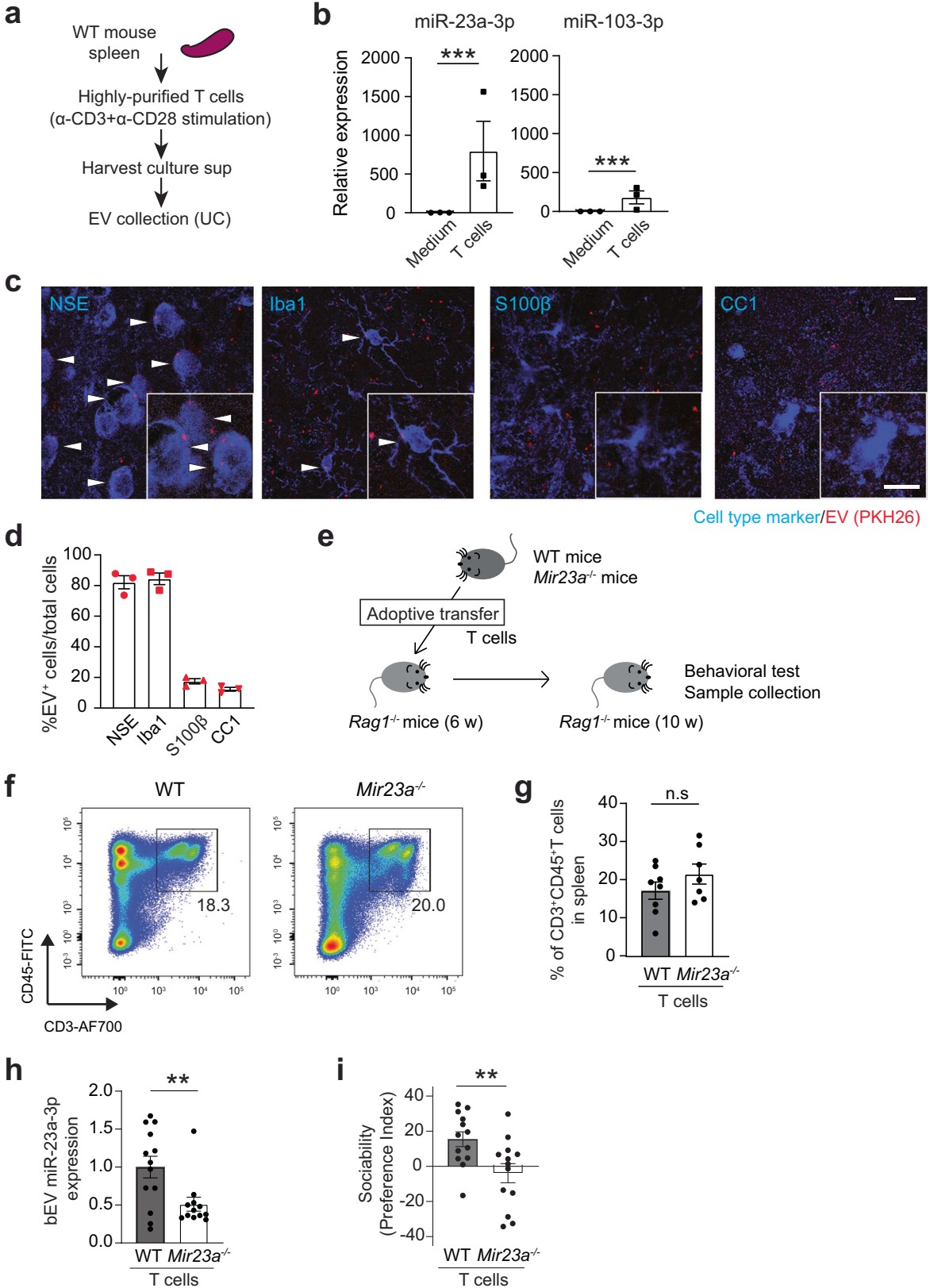

Cell type marker/EV (PKH26)

Fos (1:500, #sc-52, Santa Cruz Biotechnology, RRID: AB_2106783), rabbit anti-c-Fos (1:200, #2250, Cell Signaling, RRID: AB_22472111), mouse anti-CaMKIIα (1:200, #688602, Clone 6G9, BioLegend, RRID: AB_2617027), rabbit anti-CaMKIIα (1:100, #20666-1-AP, Proteintech, RRID:AB_2878722), rabbit anti-RFP (1:1000, #600-401-379, Rockland, RRID: AB_2209751), rabbit anti-PKCε (1:100, #20877-1-AP, Proteintech, RRID:AB_10697812), mouse anti-CC1 (1:100, #OP80, Calbiochem, RRID:AB_2057371), rabbit anti-S100β (1:200, #ab868, Abcam, RRID:AB_306716), mouse anti-NSE (1:500, #66150-1-g, Proteintech, RRID:AB_2881546), mouse anti-Gephyrin (1:100, #147021, Synaptic Systems, RRID:AB_2232546), and rabbit anti-GABA$_A$Rγ2 (1:100, #AGA-005, Alomone Labs, RRID: AB_2039870). After washing with PBS, the

**Fig. 8 | T cell-derived miR-23a-3p contributes to circulating bEVs and sociability. a** Experimental outline of splenocyte collection, primary T cell culture, and subsequent EV collection from T cell culture supernatants. **b** Levels of miR-23a-3p and miR-103-3p expression in T cell-derived EVs ($n = 3$ biologically independent experiments). **c** Representative confocal microscope images of PKH26-labeled T cell-derived EVs (red) colocalized with neurons (NSE⁺ cells) and microglia (Iba1⁺ cells), but not with astrocytes (S100β⁺ cells) or oligodendrocytes (CC1⁺ cells), in the mPFC of *Rag1⁻/⁻* mice. EV⁺ cells are indicated by white arrowheads. Scale bars, 10 μm. **d** Quantification data of T cell-derived EV co-localization with specific brain cell types ($n = 3$ mice). **e** Experimental outline of T cell adoptive transfer and subsequent behavioral assay. **f** Representative flow cytometry data showing the

comparable reconstitution of T cells in *Rag1⁻/⁻* mice by adoptively transferred WT and *Mir23a⁻/⁻* T cells. See Supplementary Data 15 for gating strategy. **g** Quantification of WT and *Mir23a⁻/⁻* T cell reconstitution in *Rag1⁻/⁻* mice ($n = 8$ for WT and $n = 7$ for *Mir23a⁻/⁻* T cell reconstitution group). **h** miR-23a-3p levels in bEVs from *Rag1⁻/⁻* mice receiving WT and *Mir23a⁻/⁻* T cells ($n = 13$ mice per group). **i** Sociability of *Rag1⁻/⁻* mice receiving WT and *Mir23a⁻/⁻* T cells ($n = 13$ each). Each bar graph represents mean ± SEM. Each dot represents one mouse. \**p < 0.05*, \*\**p < 0.01*, and \*\*\**p < 0.005*. n.s., not significant. Significance was determined by Student's *t*-test. See Supplementary Data 13 for details of the statistical analysis. Source data are provided as a Source Data file.

sections were further incubated with fluorophore-conjugated secondary antibodies at 1:400 dilution for 2 h at room temperature, followed by DAPI staining (1:50,000, #10236276001, Roche) for 10 min at room temperature. The sections were mounted on glass slides with Permafluor™ mounting medium or ProLong Diamond antifade mounting medium (Thermo Fisher Scientific). Images were acquired using Zeiss LSM510, 700 and 800 confocal microscopes with Zen software (Carl Zeiss) or an Olympus BX61 epifluorescence microscope (Olympus).

### EV labeling and brain distribution analysis

For chemical dye labeling of EVs, WT EVs were stained with PKH26 lipophilic dye (Sigma-Aldrich). Briefly, EVs were suspended with 194 μl of Diluent C, mixed with 6 μl of PKH26 dye, and incubated for 5 min at room temperature. The reaction was stopped by adding 100 μl of 20% BSA. Then, the EV suspensions were applied to the Exosome Spin Column (Thermo Fisher Scientific) and spun down to remove excess amount of the dye. Next, stained EVs were diluted to $2 \times 10^{10}$/mL with DMEM medium (Thermo Fisher Scientific). For genetically labeling of EVs, EVs were collected from the sera of mTmG mice, in which cellular membranes and thus cell-derived EVs are labeled with membrane-anchored tdTomato. EVs ($2 \times 10^9$ particles /mouse) were intravenously injected into recipient mice. Brains were collected 1 h later and PKH26 and tdTomato⁺ puncta were analyzed by immunohistochemistry.

### Image analysis

Image analysis was performed as previously described[60]. c-Fos quantification was conducted using Image J/Fiji software[71] and the percentages of c-Fos-expressing cells among NeuN neurons were calculated. Images were taken at 20× and 40× magnification. The maximal projection images of z-stacked images (14 sections at 0.75 μm) were used for quantification with confocal. Three to five brain sections from each animal were chosen based on anatomic landmarks to ensure that equivalent regions were analyzed. EV co-localization images with each cell type marker (NSE⁺, Iba1⁺, S100β⁺, and CC1⁺ cells) were taken with 63× magnification. The maximal projection images of z-stacked images (14 sections at 0.34 μm) were used for quantification. The co-localization was defined when at least one EV puncta were overlapped with the cell marker staining. The percentages of EV-positive cells among all the cells for each cell type was quantified per visual field and averaged across 4−5 independent fields to calculate %EV-positive cells/total cells. The number of EVs was measured using ImageJ software as follows. Images of 1024 × 1024 pixels were taken with 63× magnification and converted to 8-bit monochromes using "Split Channel" function. The number of EVs per field was calculated by counting particles between 3.5 and 10 pixels² using the "Analyze Particles" function, after setting the threshold to −50. For synaptic GABA receptor quantification, images were taken using a 63× magnification objective with 2× optical zoom. The maximal projection images of z-stacked images (14 sections at 0.34 μm) were used for quantification. Postsynaptic GABA_AR γ2 was determined by the co-localized puncta of GABA_AR γ2 and gephyrin on the soma of CaMKIIα⁺ neurons. The

number of GABA_AR γ2⁺ gephyrin⁺ puncta per CaMKIIα⁺ neurons was counted and averaged across 5-6 CaMKIIα⁺ neurons per mouse.

### RNA-seq

Total RNAs were extracted from frontal cortices of mice transcardially perfused with ice-cold PBS using RNeasy micro kit (Qiagen). Libraries were prepared with NEB Next Ultra II Directional RNA Library Prep kit (New England BioLabs) and a 75-bp paired-end sequencing was performed on a NextSeq500 (Illumina) at the UAB Genomics Core. Raw RNA sequencing reads were aligned to the mouse reference genome (GRCm38 p6, Release M24) from Gencode with STAR (version 2.7.5c) (using parameters -outReadsUnmapped Fastx -outSAMtype BAM SortedByCoordinate -outSAMattributes All)[72]. Following alignment, HTSeq-count (version 0.12.3) was used to count the number of reads mapping to each gene (using parameters -r pos -t exon -i gene_id -a 10 -s no -f bam)[73]. Normalization and differential expression were then applied to the count files using DESeq2 (version 1.28.1) following their vignette[74]. Lists were compiled of genes whose expression was significantly altered between *Rag1⁻/⁻* mice and WT mice, and between *Rag1⁻/⁻* mice with and without WT EVs. *P* values were adjusted using the Benjamini-Hochberg false discovery rate (FDR) and the adjusted $p < 0.05$ was considered statistically significant. Then, genes whose expression changes were overlapped were extracted for the downstream analysis. Gene ontology (GO) enrichment analysis was performed using Metascape[75] and PANTHER, including biological process (BP), molecular function (MF) and cellular component (CC).

### miRNA-seq

Total RNAs containing small RNA fractions were extracted from EVs enriched with differential ultracentrifugation using miRNeasy mini kit (Qiagen). Libraries were prepared with Qiagen miRNA library prep kit (Qiagen) and a 75-bp single-end sequencing was performed on a NextSeq500 (Illumina) at the UAB Genomics Core. Raw miRNA-Seq FASTQ reads were uploaded to Qiagen's GeneGlobe Data Analysis Center (https://geneglobe.qiagen.com/us/ analyze/) for analysis. Briefly, the reads were trimmed, unique molecular identifier (UMI) sequences identified, and then aligned to miRbase (ver21)[76] and the mouse mm10 genome. This created a tab-delimited file containing the count reads and UMIs assigned to each miRNA. The UMIs were then normalized, and differential expression was calculated using DESeq2. For miRNA over-representation analysis, we curated the previous publications (at least 2 publications for one mouse model) and established the altered miRNA lists from several mouse models with sociability impairment[77–101]. The lists of miRNAs were converted to ver21 with miRBaseConverter[102]. Enrichment of altered miRNAs was analyzed with Fisher's exact test. *P* values from multiple testing were adjusted (*q*-value) using the Benjamini−Hochberg false discovery rate (FDR) with a significant level of 0.05. For pathway analysis of potential target genes of miRNAs, target genes of miRNAs were obtained with miRTarBase and TargetScan 8.0[103,104]. GO enrichment analysis of miRNA target genes was performed using Metascape as described above.

## Brain slice electrophysiology

Adult mice were anesthetized through isoflurane inhalation and then decapitated. The brain was quickly removed and coronal mPFC slices of 300 μm were obtained using a Leica VT1200S vibratome. The dissection buffer used for the slicing was at pH 7.3 and 305 mOsm, containing the following (in mM): 206 sucrose, 25 NaHCO$_3$, 2.5 KCl, 10 MgSO$_4$, 1.45 NaH$_2$PO$_4$, 0.5 CaCl$_2$, and 11 d-glucose. This buffer was kept oxygenated (5% CO$_2$–95% O$_2$) during the whole dissection. Slices were then transferred to a holding chamber containing artificial cerebrospinal fluid solution (aCSF), composed of the following (in mM): 126 NaCl, 26 NaHCO$_3$, 2.5 KCl, 1.45 NaH$_2$PO$_4$, 1 MgCl$_2$, 2 CaCl$_2$, and 9 d-glucose. Slices were kept oxygenated at room temperature (22–25 °C) for at least 1 h. After this resting period, they were transferred to a submersion-type recording chamber upon a modified microscope stage, and kept constantly perfused with oxygenated aCSF throughout the experiments. The temperature in the recording chamber was kept at 30 ± 1.5 °C using an inline heater (Warner Instruments). Recording electrodes pipettes were fabricated from borosilicate glass with input resistances of ~4–7 MΩ. We used a visualized slice setup under a differential interference contrast-equipped microscope to perform our electrophysiology experiments. Whole-cell patch-clamp recordings were made in neurons visually located in layers 2/3 and 5 of the prelimbic region of the mPFC. Spontaneous postsynaptic currents were recorded in voltage-clamp mode by using the SutterPatch double IPA system and SutterPatch software. We applied holding at −70 mV, and compensated the electrode and cell capacitance. For all experiments, recordings were taken only 5 min after the whole-cell configuration was achieved, to allow cell adaptation. Currents were recorded for 5 min for all experiments. We included in our data only recordings from cells in which the access resistance did not change more than 20% during the recording and that displayed relatively stable baselines. Synaptic currents were detected and measured using SutterPatch software in-built system for analysis, and average amplitude e frequency of events for each cell was taken for statistical analysis. For sEPSC experiments, the GABA$_A$ antagonist bicuculline at 20 μM plus the NMDA antagonist D-APV at 50 μM were added to the bath to isolate AMPA currents. The intracellular solution used was K$^+$ based and contained the following (in mM): 117 K-gluconate, 10 HEPES, 2 Na$_2$ATP, 0.4 Na$_2$GTP, 1 MgCl$_2$, 0.1 EGTA, 13 KCl, 0.07 CaCl$_2$, at pH 7.3, and 290 mOsm. For sIPSC experiments, the AMPA antagonist DNQX at 20 μM plus the NMDA antagonist D-APV at 50 μM were added to the bath, to isolate GABAergic currents. The intracellular solution used was Cs based and contained the following (in mM): 140 CsCl, 10 HEPES, 2 Na$_2$ATP, 0.4 Na$_2$GTP, 1 MgCl$_2$, 0.05 EGTA, 3.6 NaCl, at pH 7.3, and 290 mOsm. For the experiments where EVs were used, we added the EVs directly to the bath at a concentration of 2 × 10$^7$, and allowed the slices to incubate for at least 1 h before starting the experiments. Recordings were taken while constantly perfused by this aCSF with the EVs added.

## RNase and detergent treatment of EVs

EVs were mixed with or without 1 μl RNaseA (ThermoFisher, EN0531) and 10 μl Triton X-100 (Sigma, T2199) and adjusted up to 200 μl by PBS. All samples were then incubated at 37 °C for 30 min, followed by heat inactivation of RNase at 95 °C for 10 min. After samples had cooled, RNA was extracted using miRNEasy mini kit (Qiagen) according to the manufacturer's protocol.

## Quantitative reverse-transcription PCR (qRT-PCR) of miRNAs

EV-derived RNA (30 ng) was used to synthesize cDNA using miRCURY LNA RT kit (Qiagen). The cDNAs were then diluted at 1:30 and qPCR was performed using 2× miRCURY SYBR Green master mix (Qiagen) and miRCURY LNA miRNA PCR Assays (#339306, YP00204772 for miR-23a-3p and #339306, YP00204063 for miR-103-3p, Qiagen) with 95 °C for 2 min, 50 cycles of 95 °C for 10 s, and 56 °C for 60 s, on a

C1000 Touch Thermal Cycler with CFX96 Real Time PCR Detection System (Bio-Rad). Cycle threshold (Ct) values were averaged across triplicates per experiment and per target. ΔCt was calculated to compare the data between groups.

## EV small RNA extraction and supplementation to other EVs

Small RNAs were extracted from EVs using miRNEasy mini kit (#217004, Qiagen). The extracted RNAs (25 ng) were mixed with 2 × 10$^9$ EV particles using Exo-Fect Exosome transfection kit. EVs loaded with small RNAs were then reconstituted with 100 μl of DMEM media and transferred intravenously into recipient mice via retro-orbital injection.

## EV loading with miR-23a-3p and miR-103a-3p mimics

miRNA mimic of 23a-3p (250 pmol, #339173 YM00470983-ADA, Qiagen) and 103a-3p (250 pmol, ##339173 YM 00470828-ADA, Qiagen) or negative control (500 pmol, #339173 YM00479902-ADA, Qiagen) were mixed with 2 × 10$^9$ EV particles using Exo-Fect Exosome transfection kit (#EXFT20A-1, System Biosciences) according to the manufacturer's instruction. EVs containing miRNAs or negative controls were reconstituted with 100 μl of Dulbecco's Modified Eagle Medium (#11965-092, Gibco) and then injected intravenously into recipient mice via retro-orbital injection.

## Statistical analysis

Normality and homogeneity of variances were initially assessed for all data sets. Data were analyzed with unpaired Student's $t$ test, and one-way ANOVA, using GraphPad Prism 7 (GraphPad Software), SPSS and R. Other analyses such as Kruskal–Wallis and Linear Regression were used when appropriate. *Post hoc* analyses were performed using Dunnett's (one-way ANOVA), Dunn's (for Kruskal–Wallis), Games–Howell's (for Welch ANOVA), or Bonferroni adjustments. $p < 0.05$ was considered statistically significant and adjusted for multiple comparisons. All data is reported as mean ± standard error of the mean. Supplementary Data 13 provides the details of each analysis, including $F$ values, sample sizes and size effects.

## Reporting summary

Further information on research design is available in the Nature Portfolio Reporting Summary linked to this article.

# Data availability

Raw RNA sequencing data are available from NCBI Gene Expression Omnibus (GEO) database (GSE268327, GSE268328). Source data are provided with this paper.

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

## Acknowledgements

We thank J. Kang, D. Mallah, D. Medeiros, B. Lo, S. Gonzalez, T. Wan, S. Kou, A. Murata, Y. Kobayashi, E. Mallick, D. Muth, D. Wilson, J. Green, H. Morishita, and T. Akimoto for technical help and advice; R. Dahl for *Mir23a*$^{-/-}$ mice. We also thank the UAB Animal Behavioral Assessment core, the UAB Civitan International Research Center, the UAB Genomics core, the UAB High Resolution Imaging Facility core, the UAB Center for Neurodegeneration and Experimental Therapeutics, and the JHU School of Medicine Microscope Facility for shared resources. This work was made possible by support from the Johns Hopkins Medicine Discovery Fund (to S-i.K.) and the National Institutes of Health (MH093458, MH113645, and MH118492 to S-i.K.). K.W.W. and S-i.K. were supported by Allen Distinguished Investigator awards through the Paul G. Allen Frontiers Group (now Allen Family Philanthropies).

## Author contributions

S-i.K. initially conceived the project with E.D. K.M., E.D., P.A.G., J.F-O., M.M., I.B., F.A., J.B., O.F., E.Y.C., J.S., M.S.A.P., T.K., N.I., R.M., I.V.L.R., and T.I. conducted the experiments and analyzed the data. D.K.C., E.D., and S-i.K. performed RNA-seq and miRNA-seq data analyses. M.V.P., K.W.W., and M.N. provided their respective expertise related to behavioral assays, extracellular vesicle analysis, and in vivo neuronal manipulation. S-i.K. wrote the original draft manuscript with E.D., K.M., J.F-O., and P.A.G., and edited it with the inputs from co-authors.

## Competing interests

S-i.K. is named as an inventor on a pending patent application (Application No. PCT/US2025/039241) by the University of Alabama at Birmingham related to the miRNA-based recovery of neuronal function and behavior described in this paper. The remaining authors declare no competing interests.
