## [Transparent Peer Review file · Nature Communications]

Circulating extracellular vesicle microRNAs mediate immune modulation of social behavior in male mice

Corresponding Author: Dr Shin-ichi Kano

Version 0:

Reviewer comments:

Reviewer #3

(Remarks to the Author)

The authors have done an outstanding job of addressing my comments and critiques. Congrats to them on this exciting work.

Reviewer #4

(Remarks to the Author)

The revised manuscript has substantially addressed many of the concerns raised in the previous review. The authors have further added extensive new datasets, which significantly enhancing the mechanistic understanding of the study. The identification of T-cell-derived EV-associated miRNAs and their downstream signaling pathway regulating PFC synaptic activity potentially represent an important advance in understanding how peripheral immune factors influence brain function and sociability behaviors.

However, there is a concern that also raised in the previous review and remains only partially addressed in the revised manuscript.

A central conclusion of the study is that WT serum or bEVs “ameliorate sociability deficits” in Rag1^{-/-} mice and other models, as lines 67-68, lines 971-973, and lines 153-156 . Many of the behavioral data and electrophysiological analysis rely primarily on two group comparison, typically KO versus KO +EV, analyzed using Student’s t-test. While these comparisons demonstrate that EV treatment significantly increase the sociability in Rag1^{-/-} mice, the interpretation that sociability deficits were “ameliorated” or “restored” inherently implies normalization toward WT levels. To support this claim, appropriate multi-group statistical analyses, ideally using two-way ANOVA (or mixed-model ANOVA) with factors of genotype (WT vs KO) and treatment (Medium vs EV) would be required. Without these analyses, the assertion that the phenotype was “restored” or “ameliorated” may be overstated. I recommend either providing the necessary analyses to support these conclusions or tempering the strength of these statements in the results and abstract, with discussion of this limitation.

Related to this point, there is notable variability in sociability score across experiments. For example, Rag1^{-/-} mice show sociability of ~10 in Fig.1b, ~18 in Fig.1e, and ~30 Fig. 7b. While significant differences are reported within each experiment, the variability across experiments complicates interpretation of the magnitude of behavioral changes or rescue across condition. I recommend to acknowledging this limitation in the discussion section.

Reviewer #5

(Remarks to the Author)

I did not take part in the previous revisions of this manuscript. Hence, as additional independent, anonymous reviewer, I have been asked to help determining whether the reviewers’ concerns have been appropriately addressed by the revisions.

As requested by previous reviewers, the authors performed additional experiments to further explore the mechanistic basis of their previous findings.

They now identified miR-23a-3p and miR-103-3p in T-cell derived EVs as the critical mediators of the observed amelioration in sociability. The proposed mechanism for rescue of sociability entails down regulation of PKC ϵ in mPFC neurons, which is followed by increased expression of GABAA receptors in pyramidal neurons and subsequent restoration of sIPSCs in the Rag1 $^{-/-}$ mice. They also confirmed the rescue of sociability upon administration of EVs from WT mice in two additional mouse models exhibiting sociability deficits (Cntnap2 $^{-/-}$ and Shank3 $^{-/-}$ mice).

Overall, the authors addressed the reviewers' comments/concerns adequately and provided a lot of new data, which is great.

However, one downside of the proposed mechanism is that neuronal hyperexcitability in untreated Rag1 $^{-/-}$ mice is defined based on the increased co-localisation of c-FOS with CaMKii, whereas upon administration of T-cell derived EVs in Rag1 $^{-/-}$ mice, the reversal of the aberrant neuronal excitability is defined based on co-localisation of c-FOS with the neuronal pan-marker NeuN.

I wonder whether the authors examine co-localisation of cFOS-CaMKii in the rescue experiment? It would be more appropriate to report this metric in the rescue experiment.

Also, while the authors describe in detail the hypothesis-driven exploration of their transcriptomics dataset that led them to the identification of PKC ϵ as a key component of the mechanism that they propose, the pleiotropic functions of PKC ϵ (which is not exclusively involved in GABA signaling) compromises the specificity of their claim. Besides, while the beneficial effects of EVs administration were replicated in Cntnap2 $^{-/-}$ and Shank3 $^{-/-}$ mice on the behavioural and electrophysiological level (for Cntnap2 $^{-/-}$ only), the proposed molecular mechanism was validated with immunohistochemistry only in Rag1 $^{-/-}$ mice but not in the other two models. This should be at least mentioned in the discussion/limitations of the study.

Version 1:

Reviewer comments:

Reviewer #4

(Remarks to the Author)

The authors have addressed concerns by revising the manuscript and adding a limitation paragraphs to the discussion. I have no additional comments.

We appreciate our reviewers' constructive comments, which help improve our manuscript. We have revised the manuscript by updating Figures and adding/modifying descriptions. To conform to the format of Nature Communications, **Extended Data Fig.** is now described as **Supplementary Fig.**

Here are our point-by-point responses. New additions/modifications in the revised manuscript are highlighted in **blue font**.

Reviewer #3 (Remarks to the Author):

The authors have done an outstanding job of addressing my comments and critiques. Congrats to them on this exciting work.

-- We are glad to hear that. Thank you for taking the time and effort in reviewing our manuscript and constructive comments and critiques.

Reviewer #4 (Remarks to the Author):

The revised manuscript has substantially addressed many of the concerns raised in the previous review. The authors have further added extensive new datasets, which significantly enhancing the mechanistic understanding of the study. The identification of T-cell-derived EV-associated miRNAs and their downstream signaling pathway regulating PFC synaptic activity potentially represent an important advance in understanding how peripheral immune factors influence brain function and sociability behaviors.

-- Thank you very much for taking the time and effort in reviewing this manuscript. We appreciate that this reviewer has evaluated our revision positively.

However, there is a concern that also raised in the previous review and remains only partially addressed in the revised manuscript.

A central conclusion of the study is that WT serum or bEVs “ameliorate sociability deficits” in Rag1^{-/-} mice and other models, as lines 67-68, lines 971-973, and lines 153-156 . Many of the behavioral data and electrophysiological analysis rely primarily on two group comparison, typically KO versus KO +EV, analyzed using Student's t-test. While these comparisons demonstrate that EV treatment significantly increase the sociability in Rag1^{-/-} mice, the interpretation that sociability deficits were “ameliorated” or “restored” inherently implies normalization toward WT levels. To support this claim, appropriate multi-group statistical analyses, ideally using two-way ANOVA (or mixed-model ANOVA) with factors of genotype (WT vs KO) and treatment (Medium vs EV) would be required. Without these analyses, the assertion that the phenotype was “restored” or “ameliorated” may be overstated. I recommend either providing the necessary analyses to support these conclusions or tempering the strength of these statements in the results and abstract, with discussion of this imitation.

-- We agree with the reviewer that the word “ameliorate” or “restore” would require a more stringent statistical analysis. Therefore, we have changed the wording to “increase” or “enhance” from “restore” or “ameliorate” and included in our discussion the limitation of our two-group comparison strategies in the revised manuscript.

Related to this point, there is notable variability in sociability score across experiments. For example, Rag1^{-/-} mice show sociability of ~10 in Fig.1b, ~18 in Fig.1e, and ~30 Fig. 7b. While significant differences are reported within each experiment, the variability across experiments complicates interpretation of the magnitude of behavioral changes or rescue across condition. I recommend to acknowledging this limitation in the discussion section.

-- Thank you for the suggestion. Throughout this project, we experienced significant changes in the mouse housing and behavioral environments and conditions, including the lab relocation and the building construction. Still, across these different environments and conditions, Rag1 KO mice consistently showed sociability deficits, highlighting the robustness of our findings. We completely agree with the reviewer and have added discussion on this limitation in the revised manuscript, as follows:

“Notably, the behavioral rescue was also selective. WT bEVs improved sociability but not social novelty preference in Rag1^{-/-} mice. Similarly, WT serum, splenocytes, and T cells failed to improve social novelty preference. Thus, circulating T cell-derived EVs may preferentially target sociability-related neuronal circuitry, highlighting the significance of possible EV tropism for specific cell types or brain regions. One limitation of this study is the fact that the Rag1^{-/-} mouse sociability scores were variable across different experiments, which was likely caused by multiple factors, particularly the changes in mouse housing conditions and behavioral rooms, that we experienced during this study. However, our data consistently showed the improvement of sociability by WT bEVs, underscoring the beneficial effects of WT bEVs.”

Reviewer #5 (Remarks to the Author):

I did not take part in the previous revisions of this manuscript. Hence, as additional independent, anonymous reviewer, I have been asked to help determining whether the reviewers' concerns have been appropriately addressed by the revisions.

As requested by previous reviewers, the authors performed additional experiments to further explore the mechanistic basis of their previous findings.

They now identified miR-23a-3p and miR-103-3p in T-cell derived EVs as the critical mediators of the observed amelioration in sociability. The proposed mechanism for rescue of sociability entails down regulation of PKC ϵ in mPFC neurons, which is followed by increased expression of GABAA receptors in pyramidal neurons and subsequent restoration of sIPSCs in the Rag1^{-/-} mice. They also confirmed the rescue of sociability upon administration of EVs from WT mice in two additional mouse models exhibiting sociability deficits (Cntnap2^{-/-} and Shank3^{-/-} mice).

Overall, the authors addressed the reviewers' comments/concerns adequately and provided a lot of new data, which is great.

-- We appreciate this reviewer's positive comments on our revised manuscript.

However, one downside of the proposed mechanism is that neuronal hyperexcitability in untreated Rag1^{-/-} mice is defined based on the increased co-localisation of c-FOS with CaMKii, whereas upon administration of T-cell derived EVs in Rag1^{-/-} mice, the reversal of the aberrant neuronal excitability is defined based on co-localisation of c-FOS with the neuronal pan-marker NeuN.

I wonder whether the authors examine co-localisation of cFOS-CaMKii in the rescue experiment? It would be more appropriate to report this metric in the rescue experiment.

-- Thank you very much for your suggestion to use the same metric for both the untreated and rescue experiments (**Fig. 4** and **7**). We have realized that the presentation of **Fig. 4a-c** was unclear and confusing. In this work, we used the same metric of c-Fos+NeuN+ cells for these experiments (**Fig. 4a, b; Fig. 7c**). **Fig. 4b** quantification was indeed that of c-Fos+NeuN+ cells, not c-Fos+ cells. **Fig. 4c** graphs for c-Fos+CaMKII+ cells were linked to the images in **Extended**

Data Fig. 7 (now **Supplementary Fig. 7a**), where we confirmed most c-Fos signals co-localized to CaMKII+ cells as we usually expect in c-Fos staining.

-- Accordingly, we have revised these figures as follows. In **Fig. 4**, we present both c-Fos and NeuN/c-Fos images for **Fig. 4a** and **4e** as we have already done for **Fig. 7c**. **Fig. 4c** data have been moved to **Supplementary Fig. 7** as **Supplementary Fig. 7b**.

Also, while the authors describe in detail the hypothesis-driven exploration of their transcriptomics dataset that led them to the identification of PKC ϵ as a key component of the mechanism that they propose, the pleiotropic functions of PKC ϵ (which is not exclusively involved in GABA signaling) compromises the specificity of their claim. Besides, while the beneficial effects of EVs administration were replicated in *Cntnap2*^{-/-} and *Shank3*^{-/-} mice on the behavioural and electrophysiological level (for *Cntnap2*^{-/-} only), the proposed molecular mechanism was validated with immunohistochemistry only in *Rag1*^{-/-} mice but not in the other two models. This should be at least mentioned in the discussion/limitations of the study.

-- We agree with the reviewer and have added further descriptions to acknowledge these points in the revised discussion as follows:

“PKC ϵ has been shown to downregulate synaptic GABAA receptors via the N-ethylmaleimide-sensitive factor and reduce inhibitory postsynaptic currents in the hippocampus⁴². Our study shows that PKC ϵ -dependent modulation of GABAergic signaling is also critical in the mPFC, where its dysregulation leads to neuronal hyperexcitability. This modulation is further regulated by circulating EV-associated miRNAs, miR-23a-3p and miR-103-3p. Considering that blood EVs are also detected in the hippocampus and cerebellum, similar EV miRNA-mediated modulation of synaptic neurotransmission may occur in other brain regions. Previous studies have reported that local protein translation controls the expression of GABAA receptors and some PKC isoforms⁴⁶⁻⁴⁸; thus, EV miRNAs may directly influence the local translation of PKC ϵ at synapses, regulating synaptic GABAA receptor level. It remains unclear whether and how the expression changes in a pleiotropic enzyme, PKC ϵ , preferentially influence GABAergic signaling in the mPFC. As the current knowledge of predicted targets by miRNAs is still limited, we cannot completely exclude that miR-23a-3p and miR-103-3p may regulate GABAergic signaling via molecules other than PKC ϵ .”

*“Intriguingly, we observed the beneficial effects of bEVs on sociability in *Cntnap2*^{-/-} and *Shank3*^{-/-} mice as well. Nevertheless, the underlying mechanisms have not been thoroughly dissected in the current study. Although we showed that bEVs improved electrophysiological phenotypes in a miR-23a-dependent manner in *Cntnap2*^{-/-} mice, it remains unclear whether the PKC ϵ -GABAA receptor axis is similarly impaired in these mice. It is also unclear whether the beneficial effects of bEVs on *Shank3*^{-/-} mice are dependent on miR-23a-3p. Further studies are required to determine the common and selective mechanisms underlying the effects of WT bEVs on sociability across multiple mouse models.”*